# Loss of testosterone impairs anti-tumor neutrophil function

Janet L. Markman[1], Rebecca A. Porritt[1], Daiko Wakita[1], Malcolm E. Lane[1], Daisy Martinon[1], Magali Noval Rivas ● [1,2,3,4], Michael Luu[5], Edwin M. Posadas[6,7], Timothy R. Crother ● [1,2,3,4] & Moshe Arditi ● [1,2,3,4 ✉]

In men, the incidence of melanoma rises rapidly after age 50, and nearly two thirds of melanoma deaths are male. The immune system is known to play a key role in controlling the growth and spread of malignancies, but whether age- and sex-dependent changes in immune cell function account for this effect remains unknown. Here, we show that in castrated male mice, neutrophil maturation and function are impaired, leading to elevated metastatic burden in two models of melanoma. Replacement of testosterone effectively normalized the tumor burden in castrated male mice. Further, the aberrant neutrophil phenotype was also observed in prostate cancer patients receiving androgen deprivation therapy, highlighting the evolutionary conservation and clinical relevance of the phenotype. Taken together, these results provide a better understanding of the role of androgen signaling in neutrophil function and the impact of this biology on immune control of malignancies.

[1] Department of Pediatrics, Division of Infectious Diseases and Immunology, Cedars-Sinai Medical Center, Los Angeles, CA 90048, USA. [2] Department of Biomedical Sciences, Infectious and Immunologic Disease Research Center, Cedars-Sinai Medical Center, Los Angeles, CA 90048, USA. [3] Department of Biomedical Science, Research Division of Immunology, Cedars-Sinai Medical Center, Los Angeles, CA 90048, USA. [4] David Geffen School of Medicine, University of California, Los Angeles, CA 90095, USA. [5] Biostatistics and Bioinformatics Core, Cedars-Sinai Medical Center, Los Angeles, CA 90048, USA. [6] Urologic Oncology Program/Uro-Oncology Research Laboratories, Samuel Oschin Comprehensive Center Institute, Cedars-Sinai Medical Center, Los Angeles, CA 90048, USA. [7] Division of Hematology/Oncology, Department of Medicine, Cedars-Sinai Medical Center, Los Angeles, CA 90048, USA. ✉email: moshe.arditi@cshs.org

In the United States, the incidence of melanoma is rising faster than any other preventable cancer[1]. With approximately 70,000 new cases each year and over 10,000 deaths attributed to the disease, Americans lose an average of 20.4 years of potential life due to melanoma mortality[2,3]. Interestingly, the rate of malignant melanoma is similar in men and women aged 15–45 years, but after age 50, the rate in males increases dramatically[4], and nearly two-thirds of melanoma deaths are male[2]. The location of melanoma development also varies by sex, with males typically having melanoma on the trunk region of the body, whereas females more frequently develop melanoma on the legs[5] and the odds of metastasis from thin lesions (≤1 mm) are three-fold higher in men[6]. In a retrospective study analyzing metastatic patterns from patients treated at a single center in Germany, females had a higher incidence of locoregional metastasis and a lower risk for distance metastasis (67.3% in females compared with 73.8% in males)[7]. In addition, numerous studies have confirmed that sex is an independent prognostic factor even after adjustment for Breslow thickness, histological subtype, body site, age, ulceration, vascular invasion, mitotic rate, sentinel lymph node positivity, and detection and diagnostic delay[8–14]. However, the reason for the rise in melanoma incidence in aging males remains to be elucidated.

The immune system is crucial for controlling malignancies. Innate immune cells can both kill tumor cells and elicit an adaptive immune response to eradicate tumors. However, the immune system can also have a negative impact by releasing factors that lead to tumor cell proliferation, angiogenesis, and metastasis, and tumors may adapt to their environment and selectively inhibit immune responses[15]. Harnessing the ability of the immune system to detect and eliminate tumor cells is the major goal of the nascent field of immunotherapy.

Many studies have documented important roles for sex hormones in the development and function of the immune system. For example, estrogen promotes the production of type 1 interferon, and is known to affect the cycling of hematopoietic stem cells and the development of dendritic cells[16]. Similarly, androgen signaling has also been shown to play an important role in both innate and adaptive immune processes, including macrophage activation during wound healing, T and B cell development, and neutrophil production[17]. However, the extent to which the actions of these hormones underlie sex-specific vulnerabilities to disease has remained unclear. Here, we used multiple syngeneic metastatic mouse models to elucidate the underlying cause of the sexual dimorphism in tumor burden in melanoma. We show that compared with normal male mice, castrated mice have higher rates of tumor formation, related to impaired maturation and oxidative responsiveness of neutrophils. Testosterone replacement reduces tumor burden in castrated mice, and bone marrow transplant experiments demonstrate that the effect is due to androgen signaling in immune cells. We observe that prostate cancer (PCa) patients undergoing androgen deprivation therapy also exhibit impairments in neutrophil maturation and function. Taken together, these data illuminate the role of androgen signaling in neutrophil function, and the potential impact of this effect on cancer progression.

## Results

**Castration increases melanoma tumor burden in the lung**. To investigate the role of sex in melanoma, we used a syngeneic mouse B16 melanoma model, in which expression of the tumor suppressors p16Ink4a and p19Arf is lost[18], a common occurrence in human melanoma[19]. We found that female mice had increased lung tumor burden compared with male mice (Fig. 1a, b). This increased tumor burden occurred throughout the lung and not only on the surface, as confirmed by melanin content quantification and lung hematoxylin and eosin (H&E) staining (Supplementary Fig. 1a–b). However, this divergence was only observed in the experimental metastatic model, as subcutaneous tumors grew at the same rate in male and female mice (Supplementary Fig. 1c). To determine whether the elevated metastatic burden in females was related to differences in weight, we assessed B16 tumor growth in $Ldlr^{-/-}$ mice, which have a high propensity to gain weight on a high fat diet (HFD)[20]. The increased tumor burden in female mice was maintained in this model, even in the setting of diet-induced weight gain and increased cholesterol (Supplementary Fig. 1d–g).

We next performed ovariectomy and castration surgeries to determine the impact of sex hormones during this process, and found that the increased tumor burden observed in the lungs of female mice was not dependent on progesterone or estrogen, as ovariectomy did not affect tumor burden (Fig. 1c, d). However, castration of male mice four weeks prior to tumor inoculation resulted in increased tumor burden (Fig. 1e, f) and impaired survival (Fig. 1g). Similar results were also observed in a second model of melanoma, YUMM1.7 (derived from $Braf^{V600E}$ $Cdkn2a^{-/-}$ $Pten^{-/-}$ GEMM mice)[21] (Fig. 1h–j). Unlike humans, mouse adrenal glands do not synthesize androgens[22], so removal of the testes in these models should result in full loss of testosterone production. Importantly, although YUMM1.7 cells originated in a male mouse[21] and do express the androgen receptor (AR) by qPCR but not by western blot (Supplementary Fig. 1h–i), B16 cells are monosomic X[23] and do not express AR by qPCR or by western blot (Supplementary Fig. 1h–i), indicating that the effect of androgens on metastatic growth is indirect and not related to the sex of the donor cells. We further confirmed that the sex of the donor mouse was not important, as YUMM3.1 cells (bearing the human mutations BRAFV600Ewt/Cdkn2$^{-/-}$), which were derived from a female mouse, exhibited the same increased lung tumor burden following tail vein injection (Supplementary Fig. 1j–k).

**Castration impairs neutrophil maturation and function**. We next performed immune profiling of the lungs of B16 and YUMM1.7 tumor-bearing mice to determine whether the differences in metastatic burden were related to differences in immune response between sexes. As there were no differences between the means of the two models and each had the same significant findings, the average of both models is shown in Fig. 2. We observed no differences between sham males, castrated males, or females in the percent of lung-localized alveolar macrophages, interstitial macrophages, dendritic cells, NK cells, NKT cells, CD4+ T cells, or CD8+ T cells (Fig. 2a; Supplementary Fig. 2a). In agreement with these results, there were also no significant differences in the percent of CD69-expressing or CD44-expressing T or NKT cells, although the percent of IFNγ-expressing NKT cells tended to be higher in sham male mice (Fig. 2b; Supplementary Fig. 2b). To further investigate the role of the adaptive immune response in susceptibility to metastases, we implanted Rag1 KO mice, which lack B and T cells, with B16 via a tail vein injection. Enhanced tumor burden in female mice was preserved in this model (Fig. 2c, d), indicating that the sex difference is not dependent on adaptive immune function. Thus, we more closely examined innate immune cells in our B16-implanted wild type (WT) mice, revealing a decreased percentage and total cell count of CD11b$^+$Ly6G$^+$ neutrophil infiltration in both castrated and female mice (Fig. 2e, f). In addition, we observed in both castrated and female mice decreased percentages and total cell count of NK cells expressing IFNγ or CD69, which can trigger cell-mediated cytolytic activity, proliferation, and release of cytokines and granule contents[24] (Fig. 2g, h). B16 cytotoxicity assays

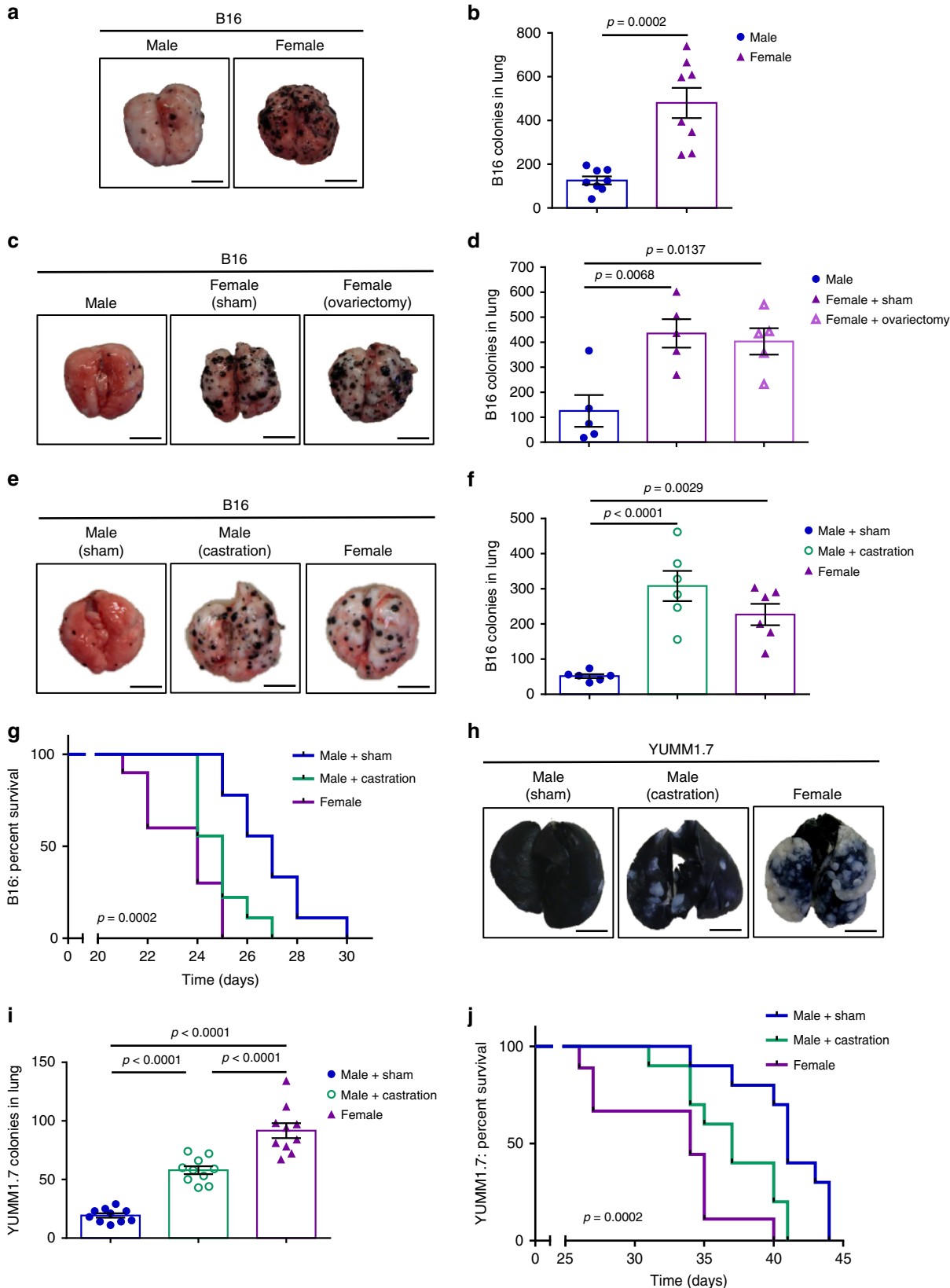

revealed a non-significant trend towards increased cytotoxicity of NK and neutrophils isolated from the lungs of male B16 tumor-bearing mice compared to castrated males and females (Fig. 2i, j). In naïve mice, a reduced percentage of neutrophils was observed in castrated males, but not in female mice (Supplementary Fig. 2c),

while no other differences in other cells types were observed (Supplementary Fig. 2d).

Since we observed a decrease in the percentage of neutrophils in castrated male and female mice compared with sham male mice (Fig. 2e), we next assessed if neutrophils contributed to the

**Fig. 1 Melanoma tumor burden in the lungs of female and castrated male mice is higher than in sham male mice in experimental metastatic models.**
Representative images of lungs (**a**) and numbers of tumor colonies (**b**) in the lungs of male and female C57Bl/6 mice following a B16 tail-vein injection ($n = 13$ per group). Representative images of lungs (**c**) and numbers of tumor colonies (**d**) in the lungs of B16-injected males, sham females, and females following ovariectomy ($n = 5$ per group). Representative images of lungs (**e**) and numbers of tumor colonies (**f**) in the lungs of B16-injected sham males, castrated males, and female mice ($n = 6$ per group). **g** Survival curve following tail-vein injection with B16 melanoma cells ($n = 9$ male+sham and male +castration; $n = 10$ female). Representative images of lungs (**h**) and tumor colonies numbers (**i**) in YUMM1.7 injected sham males, castrated males, and female mice ($n = 10$ per group). **j** Survival curves following tail-vein injections of YUMM1.7 melanoma cells ($n = 10$ male+sham and male+castration; $n = 9$ female). Data in **b**, **d**, **f**, and **i** are mean ± s.e.m. by two-tailed unpaired $t$ test (**a**) and 1-way ANOVA ith Bonferoni post-tests analysis (**d**, **f**, and **i**). Survival curve analysis (**g**, **j**) was performed using the two-sided Log-rank (Mantel-Cox) test. Data in **a–b** and **e–f** are representative of at least 3 independent experiments; data in **c–d** and **h–j** are representative of 2 independent experiments. The scale bar represents 0.5 cm for images of lung.

tumor burden differences between the sexes. Depletion of neutrophils with anti-Ly6G antibody (αLy6G) resulted in increased tumor burden in male mice, but not female mice (Fig. 3a, b; Supplementary Fig. 3a). Interestingly, neutrophil depletion also led to a significant reduction in IFNγ-producing NK cells in males (Fig. 3c), indicating cross-talk between neutrophils and NK cells. In vivo depletion of NK cells with an anti-NK1.1 antibody (αNK1.1) elicited a significant increase in tumor burden in both male and female mice, suggesting that NK cells play an important role in blocking tumor growth in both sexes (Supplementary Fig. 3b–d). Thus, although NK cells are crucial for the killing of B16 cells, NK cells do not participate in the observed sex difference in experimental metastases between male and female mice.

In an effort to understand the sex differences in neutrophil phenotypes, we analyzed their maturation and functional states. Neutrophils can be divided into subtypes based on their maturation levels[25]. Immature neutrophils include myelocytes, which mature into meta-myelocytes, and then into banded neutrophils. Neutrophils are considered mature once the nucleus becomes segmented[25]. We found that tumor-bearing sham male mice had an increased percentage of mature, segmented neutrophils compared with both castrated and female mice (Fig. 3d; Supplementary Fig. 3e). However, this difference was not observed in naïve mice, indicating that it is not an intrinsic difference (Supplementary Fig. 3f). As neutrophils mature in the bone marrow, CXCR2 expression increases, whereas the expression of CXCR4 and VLA-4 decrease[26]. Consistent with this model, neutrophils isolated from the bone marrow of tumor bearing castrated male mice expressed lower CXCR2 mRNA levels and higher CXCR4 and VLA-4 mRNA levels compared with those from sham male mice (Supplementary Fig. 3g). Similarly, female-derived neutrophils had decreased expression of CXCR2 and increased expression of VLA-4, but there was no difference in CXCR4 (Supplementary Fig. 3g). CD62L expression, which correlates with CXCR2 and inversely correlates with CXCR4[26], was also lower in neutrophils isolated from castrated and female mice compared with male mice (Supplementary Fig. 3g). Finally, STAT3 expression, which impacts the retention and release of neutrophils in the bone marrow[27], was similarly decreased in castrated and female-derived neutrophils and may underlie the reduced neutrophil chemotaxis to the lung in those groups (Supplementary Fig. 3g).

To elucidate the effects of testosterone signaling via AR in neutrophils, we performed a profiler qPCR array of AR signaling targets in isolated neutrophils from sham and castrated male mice. Of the 84 genes tested, 40 genes were upregulated, while only 4 genes were downregulated in the neutrophils from castrated males (>2-fold difference between groups), indicating that the AR signaling pathway in neutrophils is active and aberrant in castrated male mice compared with sham (Supplementary Fig. 3h). Interestingly, castration resulted in upregulation of AR in neutrophils, which has previously been shown in

prostate cancer cells treated with an AR antagonist[28]. As an additional control, we isolated prostate tissue from sham and castrated males and performed the qPCR array of AR signaling targets. In direct contrast to isolated neutrophils, and as expected, 53 AR signaling genes were downregulated in castrated males, while only 5 genes were upregulated (>2-fold difference between the group; Supplementary Fig. 3i). Validation of selected markers in neutrophils from castrated males and females (Supplementary Fig. 3j) confirmed our results.

We next assessed neutrophils' ability to generate reactive oxygen species (ROS). Oxidative burst occurs during phagocytosis and in response to soluble agonists, and while this action is important for host defense, it can also damage surrounding tissues[29]. Unstimulated neutrophils isolated from the lungs of tumor-bearing castrated and female mice had an increased percent of dihydrohodamine 123 (DHR123)-producing cells, which measures production of $H_2O_2$ (Fig. 3e; Supplementary Fig. 3k). However, upon stimulation with PMA, a significantly increased percent of sham male neutrophils produced ROS compared with neutrophils from castrated and female mice (Fig. 3f). The enhanced percent of ROS-producing male neutrophils in response to PMA was also observed in naïve mice (Supplementary Fig. 3l–m), indicating that neutrophils from castrated mice have an innate defect in responding to stimuli. In addition, neutrophils from the lungs of male B16-tumor bearing mice exhibited enhanced phagocytosis of FITC-labeled zymosan beads (Fig. 3g), again indicating a more efficient response to stimuli. We also performed immunohistochemistry to determine the distance of neutrophils within 100 μM of a B16 tumor nodule in the lung. Neutrophils in sham male mice were found to be in contact with the tumor, which is represented by a distance of 0 μM, more often (47%) than in castrated male (26%) or female mice (17%), with a significant difference in distance in sham male compared with female mice, and a similar trend in castrated mice (Fig. 3h). While we do not know the function of these neutrophils, these results warrant further investigation in future studies.

To further elucidate whether it was the hormonal environment or the sex of the immune cells that accounted for this effect, bone marrow transplants (BMT) were performed in which C57BL/6 male mice received either male (control) or female CD45.1[+]BM, and female mice received female (control) or male CD45.1[+] BM (Supplementary Fig. 3n). Male mice receiving either male or female BM still had decreased lung tumor burden (Fig. 3i–j), increased neutrophil infiltration (Fig. 3k), and an increased percent of DHR123-producing neutrophils following stimulation (Fig. 3l) compared with female recipient mice, indicating that the hormonal environment, rather than the sex of the immune cell, underlies the sex differences in immune response.

**Physiological testosterone reverses the castrated phenotype.** To determine the impact of testosterone (T) on the observed sex differences in melanoma tumor burden in the lung, 1.5 mg T

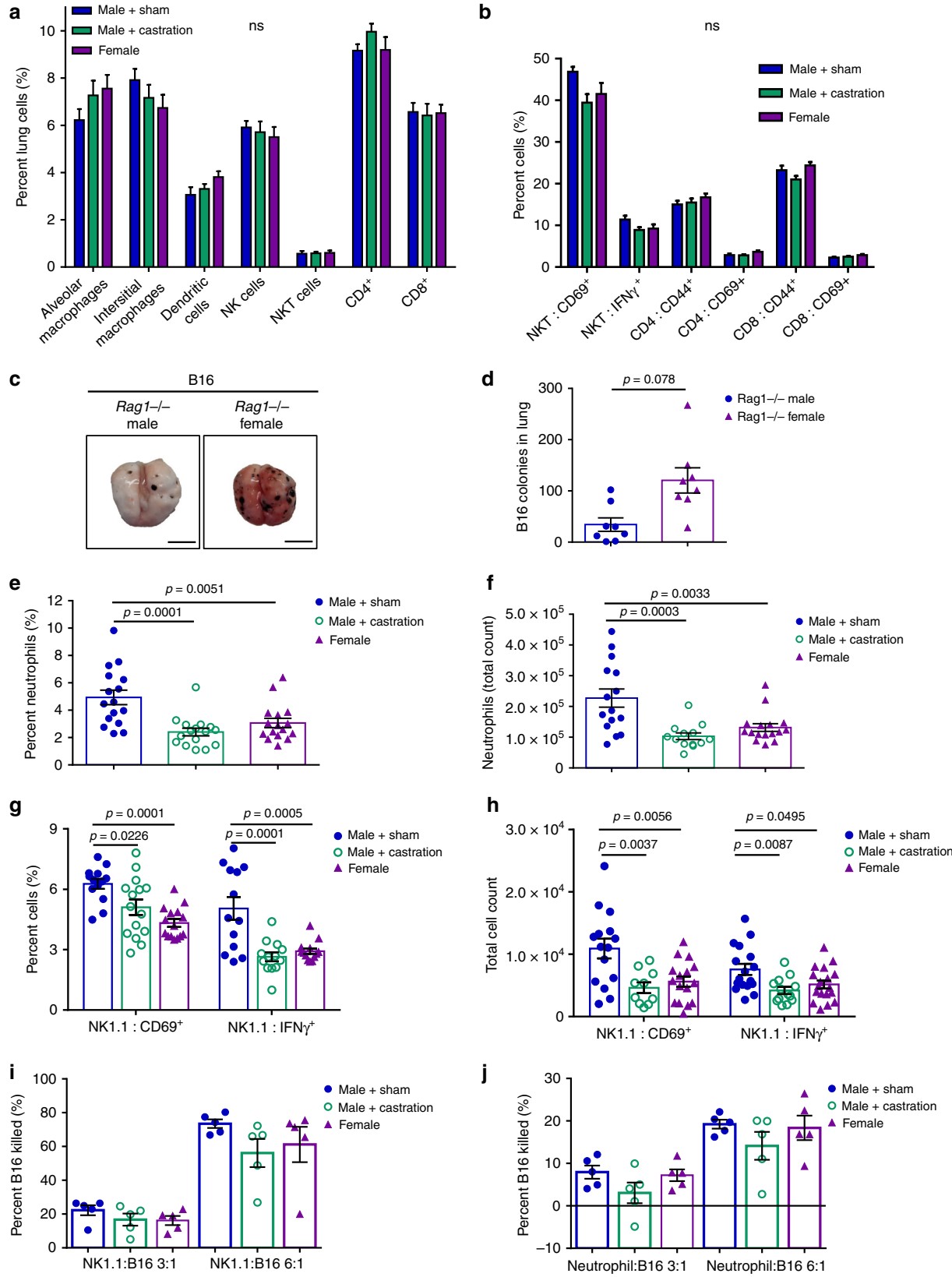

pellets were implanted at the time of castration to replicate physiological levels (Supplementary Fig. 4a). While castration led to an increase in tumor burden, testosterone replacement decreased tumor burden to the level seen in sham male mice, and neutrophil infiltration similarly increased to sham levels

(Fig. 4a–c). As androgens have been reported to be immuno-suppressive[30], we also assessed the effect of high dose T at the time of castration, and found that indeed tumor burden did increase (Supplementary Fig. 4b–c), although the percentage of DHR123-producing neutrophils following stimulation was

**Fig. 2 Sham male mice have increased neutrophil infiltration and NK cell activation in the lungs of melanoma injected mice. a** Average immune cell content in the lungs of B16 and YUMM1.7 tumor-bearing mice, based on flow cytometry for the following cell types: CD11b⁻ CD11c⁺ F4/80⁺ alveolar macrophages, CD11b⁺ CD11c⁻ F4/80⁺ interstitial macrophages, CD11b⁺ CD11c⁺ F4/80⁺ dendritic cells, NK cells, NKT cells, CD4⁺ T cells, CD8⁺ T cells (*n* = 12 male groups; *n* = 14 females). **b** Analysis of percent-activated immune cells by flow cytometry: CD69⁺ NKT cells, IFNγ⁺ NKT cells, CD4⁺CD44⁺ T cells, CD4⁺CD69⁺ T cells, CD8⁺CD44⁺ T cells, CD8⁺CD69⁺ T cells (*n* = 14 male groups; *n* = 16 females). Representative lung images (**c**) and tumor colonies numbers (**d**) in the lungs of *Rag1⁻/⁻* male and female mice following B16 tail-vein injection. **e–f** Flow cytometry analysis of lung-infiltrating CD11b⁺ Ly6G⁺ neutrophils and (**e**: *n* = 16 per group; f: *n* = 15 male+sham, *n* = 16 male+castration, female) **g–h** CD69⁺ NK cells and IFNγ⁺ NK cells (**g**: *n* = 13 male +sham, *n* = 15 male+castration, *n* = 11 female; **h**: *n* = 15 male+sham, *n* = 10 male+castration, *n* = 16 female). Ex vivo B16 cytotoxicity assays of lung isolated (**i**) NK cells and (**j**) neutrophils from tumor-bearing mice at effector cell:target cell ratios of 3:1 and 6:1. Data in **a–b**, **d–h**, and **i–j** are mean ± s.e.m. by 1-way ANOVA with Bonferoni post-tests analysis (**a–b** and **e–h**) or two-tailed unpaired *t* test (**d**). Data in **a–b** and **e–h** are representative of at least 3 independent experiments; data in **c–d** are representative of 2 independent experiments (*n* = 8 per group); data in **i–j** are representative of 2 independent experiments (each point is from 3 pooled mice for an *n* = 15 per group). The scale bar represents 0.5 cm for images of lung.

similar among sham and both T supplemental concentrations (Supplementary Fig. 4d). These data indicate that T may be beneficial to the immune response at physiologic levels, but detrimental at super-physiologic doses. Interestingly, supplementing female mice with a low-dose T pellet also caused a small decrease in tumor burden (Supplementary Fig. 4e–f). In addition, we implanted YUMM1.7 cells into the right flanks of mice and allowed the cells to spontaneously metastasize to the lung. In agreement with the subcutaneous B16 model (Supplementary Fig. 1c), there was no significant difference in primary tumor size between sham male and female mice (Fig. 4d). However, there was a non-significant trend towards an increased number of YUMM1.7 colonies in the lungs of female mice (Fig. 4e). In comparison, castrated males had both a significantly increased primary tumor volume (Fig. 4d) and increased number of YUMM1.7 metastatic colonies in the lung compared with sham and castrated males with T supplementation (Fig. 4e). While there was not a large number of metastatic colonies found in any mouse as they reached euthanasia criteria early due to the size of the subcutaneous tumors, these data do show an increased frequency and number of micro-metastases, again supporting the protective role that T has in preventing metastatic spread of melanoma to the lung.

Previously, AR deficient mice were shown to suffer from neutropenia due to reduced proliferative activity of neutrophil precursor cells[31]. In line with this, T replacement at a physiologic dose prior to tumor inoculation produced a trend towards increased circulating neutrophils in both male and female mice, whereas castration alone decreased the number of neutrophils (Supplementary Fig. 4g).

**BMT with AR−deficient mice reflects the castrated phenotype**. In testicular feminization (AR^Tfm) mice, the translation of AR is prematurely terminated due to a single nucleotide deletion of exon 1[32]. These mice appear female, have very small testes, and do not develop androgen-producing Leydig cells properly[33]. We performed a BMT with male AR^Tfm donors to Ly5.1 C57BL/6 WT male recipients to determine if the presence of AR on immune cells impacted tumor burden. Indeed, the B16 tumor foci count in lungs of mice that received AR^Tfm BM was increased compared with control C57BL/6 WT recipients (Fig. 4f–g). Surprisingly, neutrophil infiltration in the lung was not changed between the two groups (Fig. 4h). However, analysis of neutrophil function revealed an increased percentage of DHR123-producing neutrophils in unstimulated AR^Tfm recipients (Fig. 4i), but a decreased percentage when neutrophils were stimulated (Fig. 4j). In addition, only 35.1% of isolated neutrophils from naive AR^Tfm mice were morphologically segmented, mature neutrophils (Supplementary Fig. 4h), compared with greater than 80% in WT male mice (Supplementary Fig. 3f). These data indicate that the maturation and function of neutrophils is impacted by the

presence of AR on neutrophils, and loss of AR decreases their anti-tumor potential.

**Anti-androgen treatment results in increased tumor burden**. In order to confirm our castration data, we treated mice with flutamide, a non-steroidal anti-androgen that works by directly binding AR to prevent testosterone signaling[34]. We subcutaneously implanted both male and female mice with either a placebo pellet or a 100 mg, 21 day release flutamide pellet one week prior to injecting B16 cells in the tail vein. Male mice receiving the flutamide pellet exhibited increased tumor burden (Fig. 4k–l) and a decreased percentage of neutrophils in the lungs (Supplementary Fig. 4i) compared with male mice implanted with placebo pellets. Female mice with flutamide pellets also had increased tumor burden compared with female mice bearing placebo pellets (Supplementary Fig. 4j–k These data further confirm that a lack of testosterone signaling increases tumor burden in the lung in an experimental metastasis model of melanoma.

**ADT-treated PCa patients exhibit neutrophil impairments**. In order to investigate if T plays a role in human neutrophil biology, we isolated neutrophils from prostate cancer (PCa) patients undergoing androgen deprivation therapy (ADT), or not currently undergoing treatment (controls) (Supplementary Table 1). Morphological analysis of circulating patient neutrophils revealed a decreased number of segmented, mature neutrophils in ADT patients (Fig. 5a; Supplementary Fig. 5a). Immune markers including CD16 have been shown to be crucial for antibody-dependent cellular cytotoxicity for the killing of both primary cancers and cancer cell lines[35]. Indeed, the presence of CD16⁺ infiltrating myeloid cells is associated with increased survival in patients with colorectal cancer[36]. Further, a significant decrease in CD16 expression on infiltrating neutrophils was shown in invasive cervical cancer compared with low grade lesions[37]. We observed a similar decrease in CD16^hi neutrophils in ADT-treated prostate cancer patients (Fig. 5b; Supplementary Fig. 5b). A trend towards increased myeloid-derived suppressor cells (MDSC)-like neutrophils, defined as HLA-DR⁻CD14⁻CD33⁺CD15⁺[38], was also observed in ADT treated patients (Fig. 5c; Supplementary Fig. 5c). We did not find any difference in the invasion ability of the neutrophils between the two groups (Fig. 5d; Supplementary Fig. 5d). However, consistent with the mouse data, an increased percentage of human neutrophils from patients on ADT exhibited production of ROS (DHR123) at baseline (Fig. 5e; Supplementary Fig. 5e) but these cells had impaired response to PMA (Fig. 5f). There was also a striking increase in the percentage of myeloperoxidase (MPO) positive neutrophils in ADT patients (Fig. 5g; Supplementary Fig. 5f). MPO production is crucial in an antimicrobial response, but enhanced levels are associated with high inflammation and

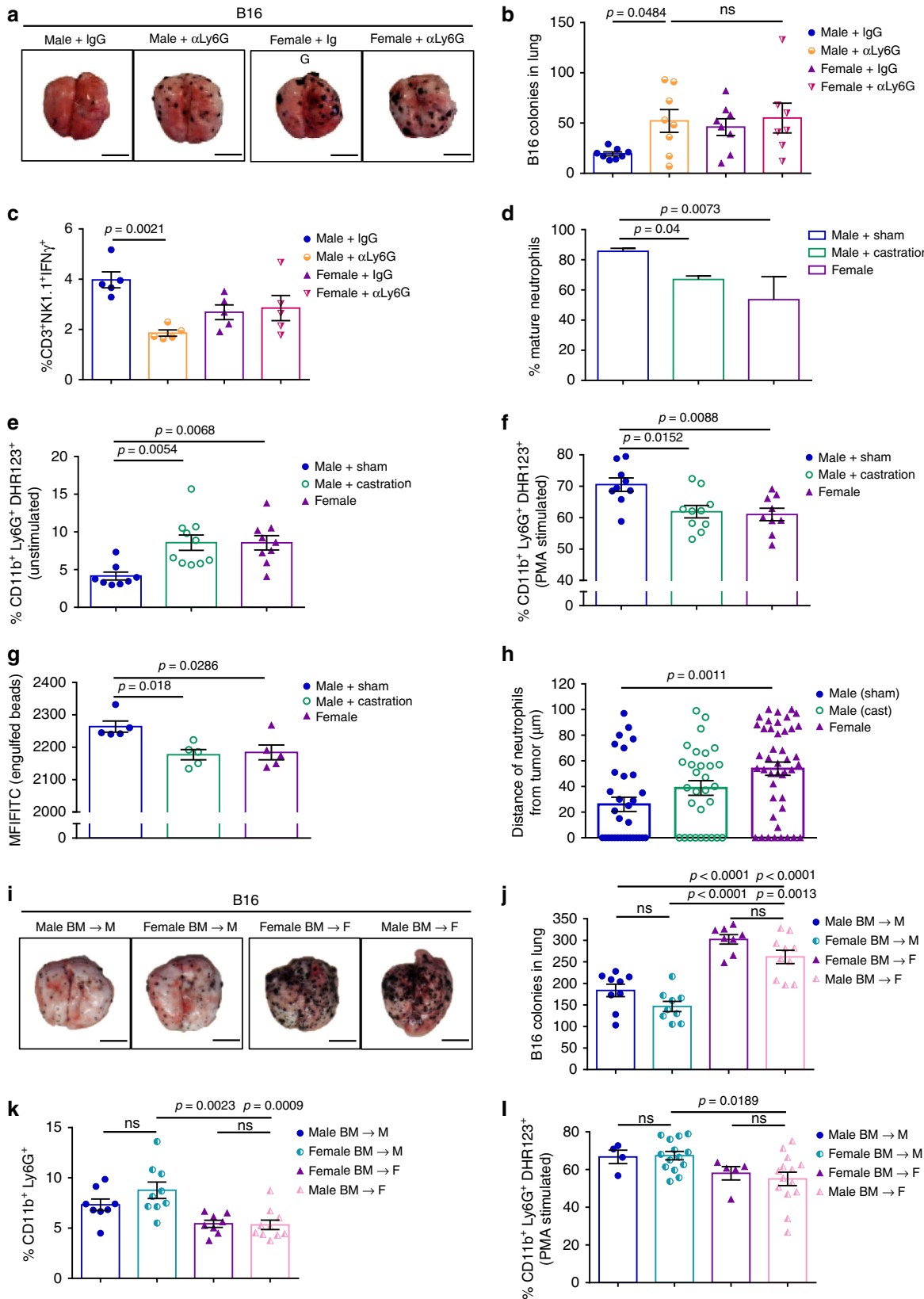

oxidative stress, and can cause DNA damage that can lead to mutagenesis[39]. MPO is also a main component of neutrophil extracellular traps (NETs). The formation of NETs is an additional important antimicrobial neutrophil function, and although the role of NETosis in cancer is still unclear, we observed that unstimulated neutrophils from ADT-treated patients produced more NETs after plating for 1 and 3 h (Fig. 5h; Supplementary Fig. 5g). Taken together, neutrophils derived from prostate

**Fig. 3 Neutrophils from sham male tumor bearing mice have an anti-tumor effect.** Representative lung images (**a**) and numbers of B16 colonies (**b**) in the lungs of male and female mice treated with either control IgG or anti-Ly6G mAb ($n = 8$ male+IgG, male+αLy6G, female+IgG, $n = 7$ female+αLy6G. **c** Percent IFNγ+ NK cells in lungs of male and female IgG control or anti-Ly6G treated mice ($n = 5$ per group). **d** Percent of lung neutrophils from tumor-bearing mice that have a mature, segmented appearance morphologically ($n = 18$ per group; 3 pooled samples each). Percentage of lung neutrophils producing DHR123 without stimulation (**e**) and after PMA stimulation (**f**) ($n = 8$ male+sham, $n = 10$ male+castration; $n = 9$ female). **g** FITC MFI of lung neutrophils after engulfing FITC-labeled zymosan A *S. cerevisiae* beads ($n = 5$ per group). **h** Distance of neutrophils within 100 μM of a B16 tumor nodule in the lung of sham male, castrated male, and female mice ($n = 30$ neutrophils/3 mice per male group, $n = 44$ neutrophils/4 female mice).Representative lung images (**i**) and number of melanoma tumor colonies (**j**) in lungs of mice receiving a BMT from the opposite sex ($n = 9$ male to F and female to M, $n = 8$ female to F, $n = 10$ male to M). **k** Percent of lung-infiltrating neutrophils following a sex BMT ($n = 8$ male to F, $n = 9$ female to M, $n = 8$ female to F, $n = 10$ male to M). **l** Percent of lung neutrophils producing DHR123 with PMA stimulation following a sex BMT ($n = 4$ male to F, $n = 14$ female to M, $n = 5$ female to F, $n = 14$ male to M). Data in **b**–**h** and **j**–**l** are mean ± s.e.m. by 1-way ANOVA with Bonferoni post-tests analysis (**b**, **c**, **e**–**h**, and **j**–**l**) or z-score by difference of two proportions (**d**). Data in **a**–**c** and **h** are representative of 2 independent experiments and data in d-g and j-l are representative of at least 3 independent experiments. The scale bar represents 0.5 cm for images of lung.

cancer patients undergoing ADT appear to be constitutively activated and may have a suppressive phenotype.

## Discussion

In this report, we utilized multiple syngeneic metastatic mouse models to elucidate the underlying cause of the well-documented male-specific increase in incidence of melanoma with age. Following injection of either the spontaneous melanoma cell line B16 or the YUMM1.7 line, which harbors human driver mutations, tumor burden was higher in female mice than male mice. While ovariectomy surgery had no impact on the formation of metastatic tumors, castration significantly increased tumor burden in the lungs of male mice. Supporting a direct role for androgen signaling in this effect, this phenotype could be completely reversed with T replacement.

It is widely accepted that sex hormones influence the development of immune cells and their responses, in a sex dependent manner. In this study, immune profiling revealed that the androgen-dependent protection from metastases was associated with differences in neutrophil maturation and function. Furthermore, neutrophil-depletion selectively impacted male mice, increasing their tumor burden and resulting in decreased NK cell activation. Previous studies have shown that the androgen receptor is important for the development of neutrophils, as AR-deficient mice have been shown to suffer from neutropenia due to reduced proliferative activity of neutrophil precursor cells[31]. We observed that castration resulted in a significantly decreased percentage of peripheral neutrophils, in agreement with previous studies[31], which could be rescued in males and increased in females via the administration of T. Thus neutrophils, which express the androgen receptor, are greatly impacted by the change in the hormonal environment in males when castration occurs. The presence of the androgen receptor on the melanoma cells was inconsequential in terms of metastatic potential, as both melanoma models exhibited the same phenotype, despite differential AR expression by qPCR, although both cell lines were negative for AR by western blot. Further, BMT experiments between sexes did not reverse this phenotype, indicating the circulating hormonal level, rather than the sex of the immune cells, underlies the effect on tumor metastasis.

The role of neutrophils in cancer remains under debate. Tumor associated neutrophils (TANs) can have potent anti-tumor effects through direct killing of tumor cells, antibody dependent cell cytotoxicity, and recruitment of other immune cells with anti-tumor activity[40–42], but have also been associated with poor prognosis[43–45] and can promote metastasis, angiogenesis, and tumor cell survival. In a UV-induced mouse model of melanoma, an UV-induced neutrophilic inflammatory immune response stimulated angiogenesis and increased migration of melanoma cells[46]. The presence of neutrophils (CD66+) and plasmacytoid

dendritic cells (CD123+) in primary melanoma patient samples have also been shown to be associated with poor prognosis[47]. Further, neutrophils have also been shown to either inhibit or promote metastasis through the formation of a pre-metastatic niche[48]. In addition, some studies have found that neutrophils may correlate with a worse melanoma outcome[46,49]. Subsequent studies have divided neutrophils into two categories: N1 and N2. N1 neutrophils are mature, non-suppressive, have pro-inflammatory anti-tumor functions, and exhibit increased expression of immune-activating cytokines and chemokines, lower levels of arginase1, enhanced cytotoxic capabilities and are capable of a potent oxidative burst[48,50]. N2 neutrophils are immature, can suppress an anti-tumor response or exhibit pro-tumor functions and support tumor growth by producing angiogenic factors and matrix-degrading enzymes, and have reduced migration, phagocytosis, and oxidative burst[48,50]. Animal studies have shown that N1 anti-tumor neutrophils in the pre-metastatic niche prevented breast cancer metastasis to the lung[40,51], while N2 immunosuppressive neutrophils in the pre-metastatic niche promoted liver cancer metastasis from the spleen[52], breast cancer metastasis (in conjunction with IL-17 producing gamma delta T cells) to the lymph nodes and lung[53], and melanoma metastasis to the lung in nude mice injected via tail vein with neutrophils and melanoma cells[54]. In the tumor microenvironment, ROS has been shown both to promote tumor development and progression, as well as cause ROS-induced apoptosis in cancer cells[48,55]. Specifically, type I interferons (IFN) have been shown to enhance the cytotoxicity of tumor associated neutrophils through increased ROS production and NET formation in mouse models, and this neutrophil activation was also shown in melanoma patients undergoing type I IFN therapies[56]. Thus, a delicate balance of ROS is required to create an anti-tumor effect, which can be further enhanced by treatment with Type I IFN[55,56]. In our studies, we observed decreased maturation of neutrophils in the blood of castrated male and female mice, which correlated with tumor burden. Changes in the RNA levels of chemokine receptors were consistent with this maturation defect, with bone-marrow derived neutrophils from tumor-bearing castrated mice characterized by decreased CXCR2 and increased CXCR4 and VLA4. Taken together, these data suggest that the presence of mature neutrophils is indicative of an N1-driven anti-tumor response.

It is likely that NK cells also play a role in this model. NK cell depletion resulted in a significant increase in tumor burden in both male and female mice in a sex-independent manner, indicating that B16 melanoma cells are very susceptible to NK cell killing. Furthermore, ex vivo studies showed increased cytotoxic capabilities of NK cells compared with neutrophils. Neutrophil-derived factors have been shown to have stimulatory effects on NK cells, including enhancement of cytotoxic activity,

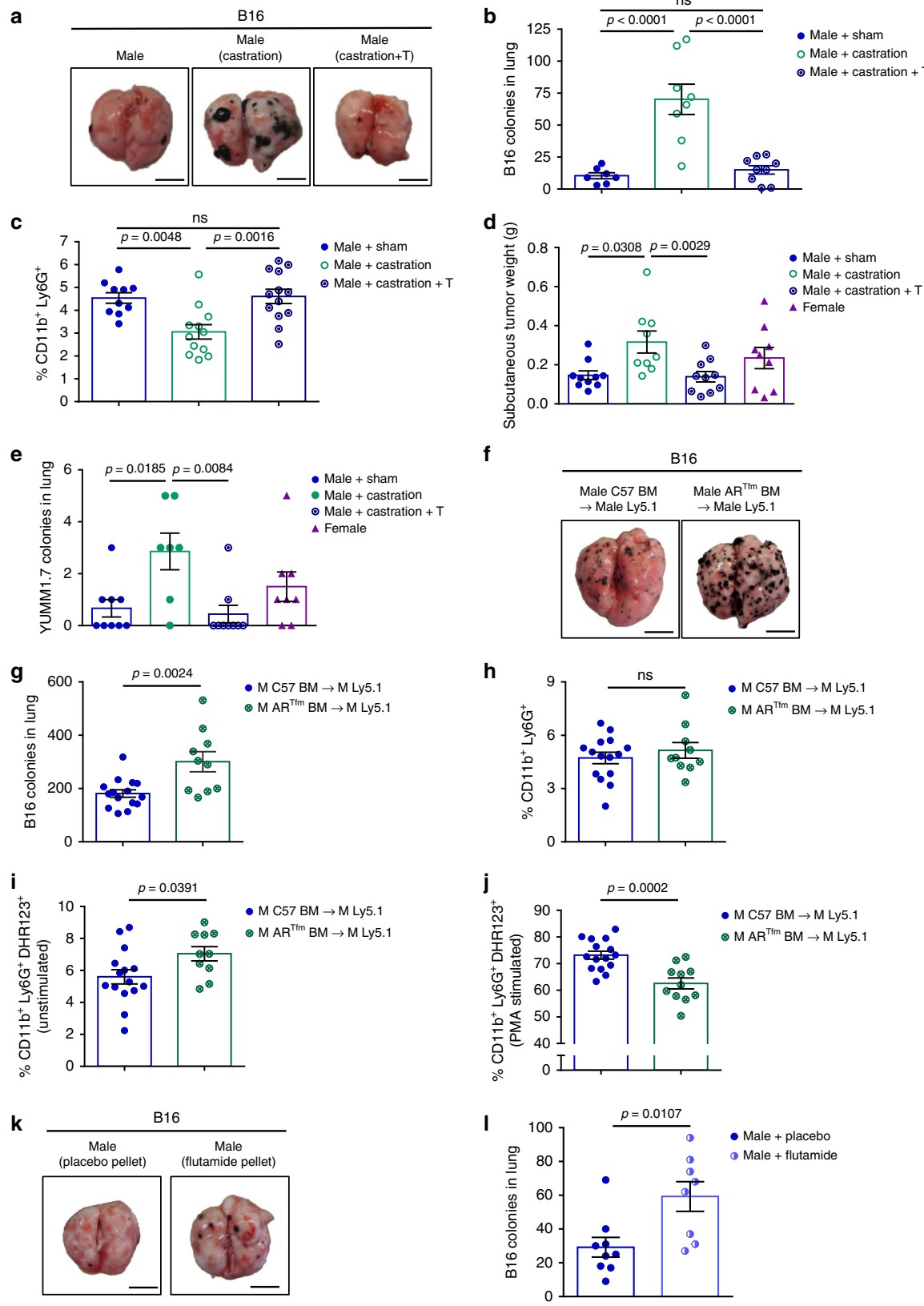

enhancement of survival, induction of CD69, and upregulation of CD11b[57]. However, granulocyte-derived ROS has inhibitory effects on a number of NK-cell effector functions, consistent with the increased percent of ROS-producing neutrophils from unstimulated castrated male-tumor and female-tumor bearing mice[57]. Further, mice specifically lacking mature neutrophils (CD11b+Ly6Ghi) exhibit NK cell hypo-proliferation and poor survival[58]. In our model, the combination of mature neutrophils

**Fig. 4 Testosterone signaling in immune cells impacts tumor burden.** Representative lung tumor images (**a**) and number of lung tumor colonies in male mice and male mice following castration with and without testosterone (T) replacement at physiological doses (**b**) ($n = 7$ male+sham, $n = 8$ male +castration, $n = 9$ female). **c** Percent of neutrophils in lung following castration with and without T replacement ($n = 10$ male+sham, $n = 12$ male +castration, $n = 13$ female). Weight of YUMM1.7 subcutaneous tumors ($n = 10$ male+sham, $n = 9$ male+castration, $n = 10$ male+castration+T, $n = 9$ female) (**d**) and number of spontaneous metastatic lung colonies ($n = 9$ male+sham, $n = 7$ male+castration, $n = 9$ male+castration+T, $n = 8$ female) (**e**). Representative lung images (**f**) and number of lung tumor colonies (**g**) following a BMT with immune cells deficient in AR (AR[Tfm] mice). Percentage of neutrophils in the lungs of male mice receiving a BMT from either WT or AR[Tfm] mice (**h**) ($n = 10-15$ per group). Percentage of DHR123-producing neutrophils following AR[Tfm] BMT without (**i**) and after PMA stimulation (**j**; $n = 15$ M C57→M Ly5.1, $n = 10$ M AR[Tfm]→M Ly5.1). Representative lung images (**k**) and number of B16 lung colonies (**l**) following implantation of either a placebo or slow-release flutamide pellet ($n = 9$ male+placebo, $n = 8$ male +flutamide). Data in **b-e**, **g-j**, and **l** are mean ± s.e.m. by 1-way ANOVA with Bonferoni post-tests (**b-e**) analysis or two-tailed unpaired $t$ test (**g-j**, **l**). Data in **a-c** are representative of at least 3 independent experiments and data in **d-l** are representative of 2 independent experiments. The scale bar represents 0.5 cm for images of lung.

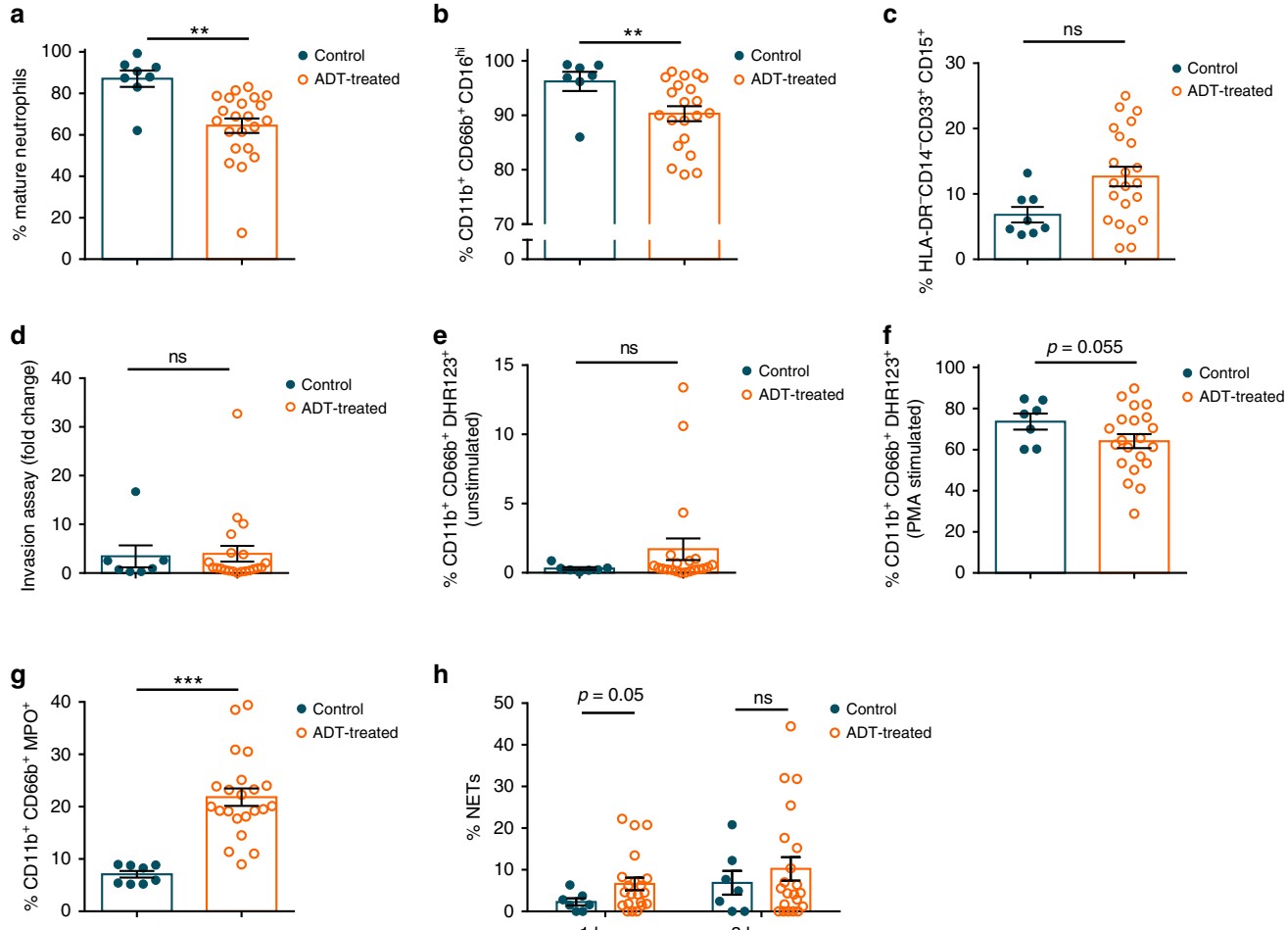

**Fig. 5 Analysis of neutrophils from control or androgen deprivation therapy (ADT) treated patients. a** Percent of circulating neutrophils from control or ADT treated prostate cancer patients that have a mature, segmented appearance morphologically. **b** Percent Cd11b+CD66b+CD16lo neutrophils. **c** Percent myeloid derived suppressor cell (MDSC)-like neutrophils defined as HLA-DR−CD14−CD33+CD15+. **d** Percent fold change in migration (percent crystal violet positive area of transwell stimulated with a chemoattractant divided by percent area without chemoattractant). **e** Percent of unstimulated neutrophils producing DHR123. **f** Percent of neutrophils producing DHR123 following PMA stimulation. **g** Percent myeloperoxidase+ (MPO+) neutrophils. **h** Percent of plated netosising neutrophils at 1 and 3 h following purification. Data in **a-h** are mean ± s.e.m., *$p < 0.05$, **$p < 0.01$, ***$p < 0.001$ by multivariable linear regression using ADT as its main predictor and adjusting for presence of prostatectomy, age, race, and current status; control prostate cancer patients ($n = 8$ in **a**, **c**, **g** and $n = 7$ in **b**, **d**, **e**, **f**, **h**) and prostate cancer patients receiving ADT ($n = 22$ in **a**, **c**, **g** and $n = 21$ in **b**, **d**, **e**, **f**, **h**).

and a decreased percent of ROS producing cells in sham male mice resulted in increased NK cell activation and IFNγ-producing cells, and ultimately, decreased lung tumor burden. While we did not find any differences in the percentage or activation of lung-derived T cells, Kissick, et al. previously showed that castration led to increased infiltration of CD3+ cells in murine lungs and decreased Th1 differentiation in both mouse and human prostate

cancer patients on ADT[59], which warrants further investigation of T cell differentiation in future studies.

We found that the addition of a physiological level of testosterone at the time of castration was able to reverse the increased tumor burden phenotype in primary tumors and both the experimental (tail vein injection) and spontaneous (subcutaneous implantation) tumor models, and that low-dose testosterone also

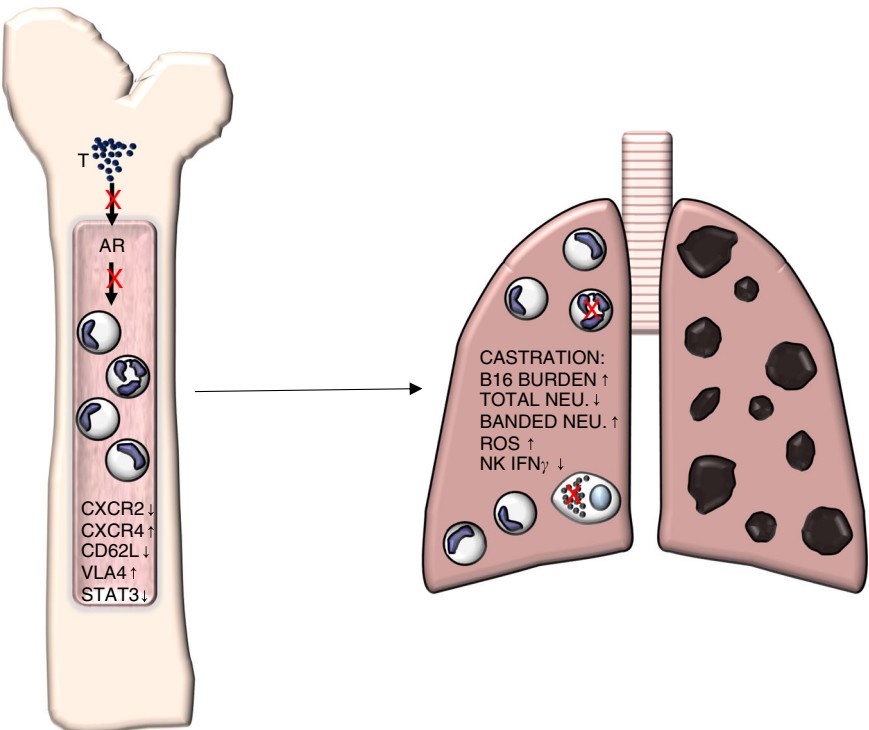

**Fig. 6 Impact of testosterone on neutrophil phenotype using a representative B16-tumor model.** In castrated male mice, testosterone is diminished and can no longer activate the androgen receptor (AR). This causes a decrease in CXCR2, CD62L, and STAT3 expression, and an increase in CXCR4 and VLA4 expression. These changes lead to more banded, immature neutrophils and an overall reduced number of neutrophils in circulation. Once B16 cells are injected via tail vein and allowed to metastasize to the lung, castrated male mice have a poor recruitment of mature, segmented neutrophils which leads to an increase of ROS-producing neutrophils, fewer NK+ IFNγ-producing cells, and results in increased tumor burden and decreased survival compared with sham-operated male mice.

had a similar protective effect in female mice following a tail vein injection with B16. Interestingly, super-physiological levels had the opposite effect, consistent with the finding that men with high T have a unique gene profile correlated with poor virus-neutralizing activity[60]. In BMTs from AR$^{Tfm}$ mice, we found that expression of truncated AR in immune cells resulted in increased B16-lung tumor burden, as well as increased ROS-producing unstimulated cells and a weakened response when stimulated. However, in this model the number of infiltrating neutrophils in the lung was consistent between groups, thus the difference in tumor burden may be explained by the decreased number of mature neutrophils, as well as the increased percent of ROS-producing cells, rather than by the number of neutrophils alone.

Taken together, these data indicate a positive effect of physiological levels of T on neutrophil function in a cancer setting. A lack of AR signaling in the bone marrow prevents the maturation of neutrophils and impedes homing to tumor-bearing lungs. Once there, these immature neutrophils exhibit a pro-tumorigenic effect with enhanced percentages of ROS producing cells, as well as suppression of NK activation and a reduced percent of NK IFNγ-producing cells (Fig. 6).

To validate that this pathway exists in humans, we isolated neutrophils from prostate cancer patients undergoing ADT treatment. ADT patient-derived neutrophils exhibited an immature, potentially MDSC-like suppressive phenotype, produced more MPO and NETs, and displayed decreased CD16 positivity. ADT is a main line of therapy for metastatic prostate cancer patients and most cancers initially respond[61]. Thus, the effect of androgen deprivation on the immunosuppressive capacity of neutrophils may not be relevant to prostate cancer progression. However, patients on ADT have an average time to failure of

11 months and an overall survival of 42 months[61]. With the recent emergence of immunotherapy for the treatment of many solid tumors, the use of checkpoint therapies for castration-resistant metastatic prostate has become increasingly intriguing. Based on these data in this study, it may be important to carefully consider how ADT can be combined with immune checkpoint blockade therapy. While some studies have shown that ADT results in an increased number of infiltrating CD3 T cells, CD4 T cells, CD8 T cells, NK cells, and macrophages in prostate tissue, as well enhanced T cell proliferation, other studies have shown that AR blockade actually impairs immunotherapy by preventing initial T-cell activation[59,62,63]. Our finding that ADT had an adverse effect on the maturity and anti-tumor potential of neutrophils, raises the question of whether ADT might further exacerbate a suppressive microenvironment and impact the efficacy of combination therapy with ADT and checkpoint blockade. Further, certain combinations of ADT with immunotherapy may exhibit better clinical efficacy, i.e., the use of androgen synthesis blockade over an AR antagonist, or the use of ADT after a positive clinical benefit with immunotherapy is already ongoing, rather than concurrent treatment.

Together, these results provide a better understanding of androgen signaling in neutrophils and the potential impact that ADT therapy in prostate cancer patients has on these cells. Further studies to understand this androgen-myeloid cell interaction are warranted, and a consideration of this phenomenon may be important when combining ADT and immunotherapy for prostate cancer patients. It may be tempting to speculate that androgen ablation therapy used in certain cancers such as prostate cancer may alter neutrophil function due to hormonal changes in the microenvironment and may increase risk of skin cancer such as melanoma, but such correlations remain to be

established in humans. In addition, while men and women initially have a similar incidence of melanoma, the risk rises in males dramatically after age 50[4], correlating with an age-dependent drop in testosterone level[64] and in line with our observations in castrated mice. While our data do not explain why males with melanoma have worse progression free survival and overall survival, our data do indicate that it may be interesting to look further at aging males, who have lower levels of testosterone, to see if hormones exert an effect on immune defense against cancer. Thus, it may be beneficial to assess hormone levels in this subset of melanoma patients and elucidate the impact that neutrophils may have on the progression of cancer. Further studies to assess function of neutrophils in patients suffering from low T are also warranted.

## Methods

**Mice.** C57BL/6, Ly5.1, Rag1$^{-/-}$, Ldlr$^{-/-}$, and B6.Cg-$A^{w-J}$ Eda$^{Ta-6J}+/+Ar^{Tfm}$/J mice were purchased from Jackson Laboratory. Age and sex matched controls were used for all of our studies. The mice were housed in a 12-h light: dark cycle at 22 °C–25 °C temperature with relative humidity of 50–70 percent, and given ad-libitum access to food and water for the duration of the study. Mice were housed in specific-pathogen-free conditions and cared for in accordance with US National Institutes of Health guidelines, and all procedures were approved by the Cedars-Sinai Animal Care and Use Committee. The number of animals used in various experiments ranged between 5 and 13 in each group, as specified in the legend of each figure.

**Cells and tumor models.** The mouse melanoma cell line B16F10 was obtained from ATCC® (CRL-6475) and was maintained in Dulbecco's Modified Eagle's Medium (DMEM) supplemented with 10% fetal bovine serum (FBS), penicillin (100 U/ml), and streptomycin (100 μg/ml). The mouse melanoma cell lines YUMM1.7 and YUMM3.1 were a generous gift from Marcus Bosenberg (Yale University, New Haven, CT). These cell lines were created from the genetically engineered mouse model of inducible melanoma harboring the following mutations: $Braf^{V600E}$; $Cdkn2a^{-/-}$; (YUMM1.7 and YUMM3.1) and $Pten^{-/-}$ (YUMM1.7 only). These mice, B6.Cg-Braftm1Mmcm Ptentm1Hwu Tg(Tyr-cre/ERT2) 13Bos/BosJ, are available for purchase at Jackson Laboratory. The cells were maintained in RPMI-1640 medium supplemented with 10% fetal bovine serum (FBS), penicillin (100 U/ml), and streptomycin (100 μg/ml).

For lung metastatic models, B16 ($2 \times 10^5$), YUMM1.7 ($5 \times 10^5$), or YUMM3.1 ($4 \times 10^5$) cells were injected via tail vein and were enumerated visually facilitated by the use of a magnifying glass and a lamp or by extracted melanin determined by spectrophotometry (for B16 only). To extract melanin, the entire left lobe was homogenized and digested in 300 μL of PBS. After centrifugation for 10 min at max speed, the supernatant was removed. The samples were then processed in 1 mL of lysis buffer containing 1 M tris-Hcl, 10%SDS, 0.5 M EDTA, 10 μg/mL Proteinase K, and water in a 56 °C shaker until completely dissolved. This process can take up to 48 h and was aided by the addition of additional proteinase K and processing through an 18-gauge needle. Once the melanin was dissolved, samples were centrifuged for 10 min at max speed and the supernatant discarded so that a black pellet remained at the bottom. 200 μL of 2 N NaOH was added and the sample placed in a shaker at 95 °C overnight or until completely re-dissolved. Then, 100 μL of 2 chloroform:1 methanol was added and the sample mixed well before being centrifuged for 10 min at max speed. 100 μL of the top layer was read on a SpectraMax spectrophotometer: OD405-OD570. Resulting values were analyzed directly. For YUMM1.7 and YUMM3.1 tumors, lungs were injected with India ink and then destained using Fekete's solution so that tumors appeared white in color. For subcutaneous tumors, $2 \times 10^5$ B16 or $2.5 \times 10^5$ YUMM1.7 cells were injected into the right flanks of C57BL mice and were measured with calipers; size was expressed as one-half the product of perpendicular length and square width in cubic millimeters. For survival studies, mice were euthanized when endpoint criteria of a Body Condition Score <2 were met. For high fat diet studies, 6-week old Ldlr$^{-/-}$ mice were placed on a Western Diet containing 21% fat and 0.2% cholesterol with 42% of calories coming from fat (Harlan Teklad, TD88137) consumed ad libitum for 12 weeks prior to inoculation of tumor cells and continued until sacrifice. Serum cholesterol levels were quantified using an enzymatic colorimetric test of total cholesterol according to the manufacturer's protocol (Cholesterol E, Wako, 439-17501).

**Histopathological analysis.** Lungs were fixed in formalin, parrafin-embedded, and sectioned. For immunohistochemistry of neutrophils, slides were stained with anti-Ly6G monoclonal antibody (mAb) (Clone 1A8, BP0075, Bio X Cell) or control rat IgG (14131, Sigma) at 1:50 overnight at 4 °C. Slides were washed with PBS 3× and stained with donkey-anti-rat IgG (AlexaFluor594, Thermofisher Scientific, A-21209) at 1:500 for 1 h at room temperature. Slides were then washed 3× with PBS and were mounted with ProLong Gold antifade reagent with DAPI (Invitrogen,

P36931). Images were obtained on a Keyence BZ-9000 microscope using ×20 magnification (Keyence). Neutrophils were identified by positive Ly6G staining and the distance from the neutrophil to the closest B16 tumor edge was calculated using the Keyence software. Neutrophils within 100 μM of a tumor nodule were included[65]. Corresponding H&E stains (MilliporeSigma, hematoxylin HHS32 and eosin 230251) of sequential tissue samples were used to confirm the location of B16 tumor nodules.

**Cell specific depletions.** In vivo neutrophil depletion was performed by intra-peritoneal (i.p.) injection of 500 μg sterile anti-Ly6G monoclonal antibody (Clone 1A8, BP0075, Bio X Cell) or control rat IgG (14131, Sigma) to both female and male C57BL/6 mice beginning one day prior to tumor inoculation and was continued to maintain depletion throughout the tumor study (days-1, 1, 4, 8, and 11). Depletion of neutrophils (Gr1$^{hi}$) in lungs was confirmed by flow cytometry at the time of sacrifice.

NK cell depletion was performed by i.p. injection of 200 μg sterile anti-NK1.1 mAb (Clone PK136, BE0036, Bio X Cell) or control mouse IgG (I5381, Sigma-Aldrich) to both female and male C57BL/6 mice beginning one day prior to tumor inoculation and was continued to maintain depletion throughout the tumor study (days-1, 1, 4, 8, and 11). Depletion of NK1.1 cells in lungs (NK1.1$^+$CD49b$^+$) was confirmed by flow cytometry at the time of sacrifice.

**Ovariectomy.** Four weeks prior to tumor inoculation, 8-weeks-old C57BL/6 mice underwent ovariectomy or sham-surgery. Mice were maintained on inhalation anesthesia (1.5% Isoflurane) via nose-cone. Prior to the start of the surgery, carprofen (5 mg/kg body weight) was administered subcutaneously. The area below the ribs was shaved and cleaned with betadine followed by alcohol. This area was then lifted with forceps to make a small 2 mm horizontal cut. Resorbable vicryl sutures were used to clamp the horn beneath the ovary and each ovary was removed using forceps and scissors. The uterine horns were then placed back into the body and the peritoneal cavity was closed using interrupted resorbable vicryl sutures. The skin was closed with interrupted nylon sutures. Following spontaneous movement, buprenorphine (0.5 mg/kg body weight) and 300 μL of warm saline were administered subcutaneously. In sham operated mice, both the skin and inner skin membrane were incised. The ovaries were externalized and returned to the abdominal cavity and the wound was sealed with interrupted nylon sutures.

**Castration and testosterone implantation.** Four weeks prior to tumor inoculation, C57BL/6 mice of 8 weeks of age were castrated or sham-operated. Mice were maintained on inhalation anesthesia (1.5% Isoflurane) via nose-cone. Prior to the start of the surgery, carprofen (5 mg/kg body weight) was administered subcutaneously. The area between the penis and the anus was shaved and cleaned with betadine followed by alcohol to disinfect the scrotum. The area between the penis and the anus were lifted to make a small 1 mm horizontal cut. To remove the testes, a small 1 mm cut into the inner skin membrane enclosing the testicles was made and the testicles were exteriorized. Testicular arteries were tied off using resorbable vicryl sutures prior to removing the testes. Once the testes were removed, the wound was sealed with two nylon sutures. Following spontaneous movement, buprenorphine (0.5 mg/kg body weight) and 300 μL of warm saline were administered subcutaneously. In sham operated mice, both the skin and inner skin membrane between the penis and anus were incised. The testes were drawn out and placed back and the wound was sealed with interrupted nylon sutures.

For hormone replacement, at the time of castration, testosterone pellets (1.5 or 50 mg dose, 60 day release, SA-151; Innovative Research of America) or placebo pellets (1.5 or 50 mg dose, 60 day release, SC-111; Innovative Research of America) were implanted subcutaneously. For the flutamide studies, 8 week old male and female mice were implanted with either a placebo pellet or a flutamide pellet (100 mg, 21 day release, A-152; Innovative Research of America) one week prior to B16 implantation. Hair from a small area behind the neck was removed with electric clippers and cleaned with betadine followed by alcohol. A small incision was made and a pellet (1 cm × 1 cm in size) containing testosterone or biodegradable matrix (placebo) were implanted subcutaneously on the lateral side of the neck between the ear and shoulder. The wound was closed with interrupted nylon sutures.

**Bone marrow transfer.** C57BL/6 or Ly5.1 mice at 8 weeks of age were lethally irradiated once with 950 rads using a $^{137}$Cesium gammacell source (Gammacell 40 Cs γ-irradiation; GammaCell Bio-Technologies). Four hours post irradiation, $1 \times 10^7$ cells were injected via tail vein under brief anesthesia while in a mouse restraint. The bone marrow cells were collected by flushing the femurs of donor mice (C57BL/6, Ar$^{Tfm}$, or Ly5.1) under sterile conditions and were prepared as a single cell suspension through a 70 μM nylon filter (BD, Falcon, 352250). The immune system was allowed to reconstitute for 8–12 weeks before tumor cells were injected. Complete bone marrow reconstitution was confirmed by flow cytometry analysis of the expression of CD45.2/Ly5.2 (from C57BL/6 mice) and CD45.1/Ly5.1 (from Ly5.1 mice).

**Complete blood count.** Fifty microliter of mouse blood was collected via retro-orbital bleeding in an EDTA coated tube and was processed the same day. A complete blood count/differential was performed by the Division of Laboratory

Medicine at UCLA (Los Angeles, CA) on a Drew Scientific HemaVet hematology analyzer.

**Testosterone level quantification**. Mouse serum was obtained at the time of sacrifice and testosterone levels were assayed using a commercially available Testosterone EIA kit (582701; Cayman Chemical) according to the manufacturer's protocol.

**Quantitative real-time PCR**. Total RNA was extracted by RNeasy extraction kit (Qiagen) and reverse transcribed using PrimeScript RT Reagent Kit (Takara Bio) according to the manufacturer's protocols. Real-time polymerase chain reaction (qPCR) was performed using SYBR Green Master Mix (Takara Bio) and was run on a CFX96 Touch Real-Time PCR Detection System (BioRad). GAPDH or beta actin served as a loading control. The following primer sequences were used for cDNA quantification: AR fwd 5'-GCCTCCGAAGTGTGGTATCC-3', rev 5'-TGG TCCCTGGTACTGTCCAA-3'; Cxcr2 fwd 5'-GGTGGGGAGTTCGTGTAGAAG3', rev 5'-CTACTACACAGGGATCAGGGC-3'; Cxcr4 fwd 5'-GCCATGGAACCGAT CAGTGTGA-3', rev 5'-GATGAAGTAGATGGTGGGCAGG-3'; L-sel (CD62L) fwd 5'-TACTGGGGGCTCGAGGAACAT-3', rev 5'-TCCCAGTTCATGGGCTTTTCA-3'; Nfkb2 fwd 5'-CGCCTCTCTTCACCTTAGGC-3', rev 5'-CAGGCCTGGATCGTAG CAAT-3'; Rab4a fwd 5'-GAGAAACCAAGAACTGCCCCT-3', rev 5'-AGCGCATT GTAGGTTTCTCG-3'; Rel fwd 5'-GAGAAACCAAGAACTGCCCCT-3', rev 5'-AA CTCCTGAAGACCTGGGCA; Stat3 fwd 5'-ATGGATGCGACCAACATCCT-3', rev 5'-CAATGGTATTGCTGCAGGTCG-3'; VLA4 fwd 5'-TTGGCTACTCGGTGGTG CTG-3', rev 5'-ATGTCTTCCCACAAGGCTCTC-3'.

**RT² profiler PCR array**. Neutrophils from lungs or prostate lobes were isolated from sham male and castrated males and were pooled with 5–6 mice per group. RNA was extracted using an miRNA extraction kit (miRNeasy, Qiagen, 217004). Synthesis of cDNA was performed using the RT² First Strand Kit (Qiagen, 330401) and cDNA was amplified using the RT² PreAMP cDNA synthesis kit (Qiagen, 330451) and AR pathway primer mix (Qiagen, PBM-142Z) for neutrophils. The RT² Profiler PCR Array Mouse Androgen Receptor Signaling Targets (Qiagen, 330231, PAMM-142ZD-2) plates were run on a CFX96 Touch Real-Time PCR Detection System (BioRad). All steps were performed according to the manufacturer's protocol.

**Western blot**. B16, YUMM1.7, and LnCaP cells (ATCC, clone FGC, CRL-1740) were pelleted and lysed in ice-cold Lysis buffer consisting of Tris (25 mM; Bio Rad; catalog 161-0719), Triton x-100 (1%; Sigma: catalog T8787), Glycerol (10%; Fisher Scientific; catalog BP229-1), EGTA (4 mM; Sigma; catalog E-4378), pH adjusted to 7.8 with dilute o-phosphoric acid. The lysates were passed 5 times through a 25-gauge needle for complete homogenization. Mouse testes were mechanically homogenized using an 18-gauge needle in ice-cold lysis buffer. Testes homogenates were then centrifuged at $10,000 \times g$ for 10 min at 4 °C to remove unbroken cells and debris. Protein quantification was performed on supernatants using bicinchoninic copper assay (ThermoFisher Scientific, catalog 23225). Equal masses of protein were loaded in 10% SDS page gels and transferred overnight at 30 V to a PVDF membrane (ThermoFisher Scientific, catalog 88518) at 4 °C. The membrane was cut below the 63 kDa and anti-AR (Santa Cruz Biotechnology, catalog sc-7305; top portion of membrane) and anti-β-Actin (Sigma, catalog A5316; bottom half of membrane) antibodies were diluted at a concentration of 1:500 and 1:10,000, respectively, in blocking buffer consisting of 5% nonfat dry milk (Bio Rad, catalog 1706404) dissolved in TBS-T. HRP-conjugated goat anti-mouse (Jackson ImmunoResearch, catalog 115-035-003) secondary antibodies were diluted at a concentration of 1:10,000 in blocking buffer. Chemiluminescent substrate (ThermoFisher Scientific, catalog 34577) was applied to membranes, and bands were imaged via Bio-Rad Chemidoc (Bio-Rad).

**Human study population**. Informed consent and blood samples were collected under Cedars-Sinai Medical Center Institutional Review Board-approved Pro00042197. For this study, a separate IRB-approved protocol (Pro00047412) was obtained to specifically study neutrophils. Patient ages ranged from 52-80 years (median 72). The patients were white ($n = 16$), black ($n = 5$), other ($n = 2$), Asian ($n = 5$), Hispanic ($n = 1$), and American Indian ($n = 1$). Patient samples were classified into two categories: patients not receiving any current treatment and patients undergoing active androgen deprivation therapy. Supplementary Table 1 indicates clinical characteristics of each patient sample used in this study.

**Neutrophil and NK cell isolation**. To isolate murine neutrophils or NK cells from the bone marrow, spleen, or lungs, an EasySep Positive PE Selection Kit (19762 A) or Negative Selection Mouse Neutrophil Enrichment Kit (18557) was used according to the manufacturer's protocols (Stemcell Technologies). The PE murine antibody used was Ly6G (1 A8; BioLegend, 127608) or NK (PK136; BioLegend, 108707). Samples were pooled from 3-5 mice for each independent experiment and purity was confirmed if >93% by flow cytometry or by Siemens Diff-Quik Stain Set (B4132-1A, Fisher Scientific).

For human neutrophil isolation, peripheral blood mononuclear cells (PBMC) were removed using SepMate tubes and the Lymphoprep density gradient according to the manufacturer's protocol (85450;07801 StemCell Technologies). Once the PBMCs were removed, the pelleted blood was transferred to a new tube and the red blood cells were lysed using 1X RBC lysis buffer (eBioscience). Cells were stained using the Siemens Diff-Quik Stain Set to confirm >94% purity. Cells were maintained in Endothelial Basal Medium-2 (EBM2, CC-3156, Lonza) which was confirmed to not activate neutrophils.

**Cytotoxicity assays**. B16 cells were plated 24 h prior to the experiment at $5 \times 10^3$ per well in a 96 well flat bottom plate. Neutrophils or NK cells were isolated from the lungs of B16 tumor-bearing mice (pooled 3 mice/sample) and were co-cultured with the B16 cells for 4 h at effector:target ratios of 3:1 and 6:1 with blank wells, effector only or target only serving as controls. All supernatant was removed and remaining adherent B16 cells were fixed with 4% paraformaldehyde overnight at 4 °C. Fixing solution was then discarded and 100 μL/well of crystal violet stain was added and plate was placed on a rocker for 30 min. Plates were then washed under running DI water for 15 min. The crystal violet was dissolved by adding 100 μL/well of 10% acetic acid with rocking and plate was read at absorbance 600 on a SpectraMax M2 spectrophotometer.

**Neutrophil scoring**. Isolated neutrophils were transferred to slides using a Cytospin 4 centrifuge (Thermo Fisher Scientific) and were stained with the Siemens Diff-Quik Stain Se (Thermo Fisher Scientific) to visualize neutrophil maturation based on nucleus morphology[25]. Each subpopulation of neutrophils was assigned an arbitrary number of 1 having a banded nuclei (immature) or 2 having a segmented nuclei (mature). Images were obtained via a Keyence BZ-9000 microscope using 40x magnification (Keyence). Greater than 150 neutrophils per sample were scored by two investigators and scores were only included if both investigators were in agreement.

**Flow cytometry**. The same lobe of the lung was manually digested in a lysis buffer containing HBSS, collagenase Type IV (2 μg/mL; 17104019; Thermo Fisher Scientific), and DNAseI (25 units/mL; 10104159001; Sigma-Aldrich) with two 10 min incubations at 37 °C. Samples were centrifuged for 5 min at 3000 RPM and the red blood cells were lysed using 1X RBC lysis buffer (eBioscience). Single cell suspensions were obtained from the digested lung and were incubated with FcBlock (CD16/CD32, Clone 2.4G2; 70-0161-M001; Tonbo Biosciences). The following murine antibodies against the respective antigens were used for flow cytometry at 1:50 concentration: neutrophils: Ly6G (1A8; PE; BioLegend, 127608) or Ly6G (Gr-1; PE; eBioscience, 12-5931-82); macrophages: CD11b (M1/70; PerCP-Cy5.5; Tonbo Biosciences, 35-0112) and F4/80 (BM8.1; APC; Tonbo Biosciences, 20-4801); dendritic cells: CD11b and CD11c (N418; vF450; Tonbo Biosciences, 75-0114); CD4 T cells: CD4 (RM4-5; APC; Tonbo Biosciences, 35-0042); CD8 T cells: CD8a (Ly-2; PerCP-Cy5.5; eBioscience, 15-0081-82); natural killer cells: NK1.1 (PK136; APC; eBioscience, 17-5941-82) or CD49b (DX5; PE; eBioscience); CD69 activation marker: CD69 (H1.2F3; PE; eBioscience, 12-0691-83); BMT confirmation: CD45.1 (A20; PerCp-Cy5.5; eBioscience, 45-0453-80) and CD45.2 (104; FITC; Tonbo Biosciences, 35-0454) and cells were fixed with 2% paraformaldehyde. For intracellular protein staining, the cells were washed and resuspended in permeabilization buffer (eBioscience, 00-5523) and stained by antibodies in the permeabilization buffer for 1 h on ice in the dark. The cells were then pelleted and resuspended in the flow cytometry staining buffer for flow cytometry analysis.

For the analysis of human neutrophils, FcR blocking antibody (130-059-901, MACS Miltenyi Biotec) was used at a concentration of 20 μL/$10^7$ cells. The following human antibodies were used: Cd11b (ICFR44; PE, Tonbo Biosciences, 50-0118-T100), CD14 (HCD14;APC/Cy7, BioLegend, 325620), CD15 (MMA; APC, eBioscience, 17-0158-42), CD16 (CB16; Alexa Fluor 700, eBioscience, 56-0168-42), CD33 (P67.6; eFluor 450, eBioscience, 48-0337-42), CD45 (HI30; violetFluor 450, Tonbo Biosciences, 75-0459-T100), CD66b (G10F5; APC, eBioscience, 17-0666-42), HLA-DR (L243; FITC, Tonbo Biosciences 35-9952-T100), Mouse IgG1 K IsoControl (P3.6.2.8.1; eFluor 450, eBioscience, 48-4714-82), and MPO (MPO455-8E6; eFluor 450, eBioscience, 48-1299-42).

For ROS staining, cells were incubated with 0.5 μg/mL dihydrorhodamine 123 (DHR123, CAS 109244-58-8; Santa Cruz Biotechnology) for 5 min with agitation at 37 °C and either remained unstimulated or were stimulated with 0.3 μM of phorbol myristate acetate (PMA) for 30 min at 37 °C. Following incubation, cells were washed and stained with CD11b and Ly6G (1A8) antibodies for murine cells or CD11b, CD45, CD66b and CD16 for human cells and were assessed by flow cytometry. Stained cells were analyzed on an LSRII Fortessa (BD Biosciences) and data were processed using FlowJo, version 10.5 (Tree Star Inc.).

**Phagocytosis assay**. Neutrophils from the lungs of B16-tumor bearing mice were plated at $2.5 \times 10^5$/well in a 24 well plate and 5 μL/well of FITC-labeled zymosan A *S. cerevisiae* beads (ThermoFisher Scientific, Z2841) were added. Following a brief spin, cells were incubated for 30 min at room temperature. The media was removed and the wells were washed with PBS. The cells were lifted by incubating for 15 min in PBS/5 mM EDTA/1 mM sodium azide plus proteinase K to cleave any beads on the surface of the cells. The cells were then stained for neutrophil markers as above and analyzed by flow cytometry.

**NETosis staining**. NETosis was induced in $2.5 \times 10^4$ neutrophils suspended in 250 µL of EBM2 media in 8-well Millicell EZ glass chamber slides (Millipore, PEZGS0816). Briefly, unstimulated cells were maintained in a 37 °C incubator for 1 and 3 h and then fixed with 4% paraformaldehyde. The slides were washed with sterile PBS three times for 5 min each and then stained with 1:2000 SYTOX orange (ThermoFisher Scientific, S11368) and 1:500 MPO (Abcam, ab9535) for 1 h at 37 °C. Slides were then washed 3× with PBS for 5 min each and then stained with 1:500 Goat pAb to Rb IgG (Abcam, ab 150077) for 1 h at room temperature. Slides were washed with PBS 3×, the chamber slide was removed according to the manufacturer's protocol and the slides were mounted with ProLong Gold antifade reagent with DAPI (Invitrogen, P36931). Four images of each time point were obtained on a Keyence BZ-9000 microscope at ×40 magnification. The total number of cells was automatically calculated with the Keyence BioAnalyzer software and the number of neutrophils was counted by a blinded scientist. The percentage of NETosing neutrophils was calculated as an average of the number of NETs divided by total number of cells/per image.

**Invasion and migration assay**. Corning transwells polyester membranes (6.5 mM, 3.0 µM pore size, Sigma Aldrich, CLS3472) were coated with human fibronectin at 10 µg/mL (Sigma Aldrich, F089S) for 1 h at 37 °C. Wells were then washed 2× with sterile PBS and dried overnight. For invasion/migration assays, the bottom chamber contained 600 EBM2 media µL with or without 100 nM N-formyl-met-leu-phe (Sigma Aldrich, 47729-10mg-f) as a chemoattractant. Neutrophils ($2.5 \times 10^4$) suspended in 200 µL of EBM2 media were placed in the upper chamber and were allowed to migrate for 2 h at 37 °C. Membranes were carefully aspirated, washed twice with PBS and stained with crystal violet for 10 min and rinsed with water until clear. Five images of each transwell were obtained on a Keyence BZ-9000 microscope at ×40 and the area of crystal violet was analyzed using the Keyence Bioanalyzer software. The results from 5 images were averaged together and a percent change in migration was calculated by dividing the percent area of transwell stimulated with a chemoattractant divided by percent area without chemoattractant for each individual patient.

**Statistical analysis**. All statistical analyses were performed using GraphPad Prism 6.0. Data are presented as mean ± SEM. For selection of appropriate statistical tests, recorded experimental data were subjected to the D'Agostino-Pearson omnibus test to detect normal, Gaussian distribution. 2-tailed Student's $t$ test or 1-way ANOVA (Tukey's post-test) were used for comparisons of parameters among two or three groups, respectively. The non-parametric Mann-Whitney U test was substituted for the student's t test when data did not follow a normal distribution. For neutrophil maturation studies, the t test for two independent proportions was used. For patient data, a multivariable linear regression analysis with adjustments for presence of prostatectomy, age, race and current status was performed by the Biostatistics Core at Cedars Sinai. White blood cell count was not available for analysis and thus was not controlled for. The value of $p < 0.05$ was considered statistically significant. $*p < 0.05$, $**p < 0.01$, and $***p < 0.001$. Statistical parameters can be found in the figure legends.

**Reporting summary**. Further information on research design is available in the Nature Research Reporting Summary linked to this article.

## Data availability
The authors declare that the data supporting the findings of this study are available within the paper and its Supplementary Information or from the authors upon reasonable request.

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

## Acknowledgements

We thank the Posadas lab staff for assistance in processing the prostate cancer patient samples, Sergio Sanders for isolation of prostate lobes, and Dr. Marcus Bosenberg for providing us with the YUMM cells.

## Author contributions

J.L.M. performed experiments and data analysis, and wrote the manuscript; R.A.P. assisted with in vivo work, data analysis, and designed qPCR primers; D.W. assisted with design of experiments, in vivo work, and data analysis; M.E.L. assisted with in vivo work and performed neutrophil staining and data analysis. D.M. assisted with Western analysis; M.N.R. contributed to the manuscript; M.L. performed the statistical analysis for the human data; E.P. participated in the prostate cancer patient procurement and completion of the IRB study; T.R.C. assisted with data analysis and editing of the manuscript; M.A. contributed ideas, edited the manuscript, and supervised this study. This work was supported by NIH R21 AI126368 (M.A.).

## Competing interests

The authors declare no competing interests.
