## [Peer Review File · Nature Communications]

Reviewers' comments:

Reviewer #1 (Remarks to the Author):

Androgens/androgen receptor signaling have a great influence on the development and functions of immune system. Previous studies in genetic mouse models have demonstrated that androgen overall induce immune-suppression and that Ar deficiency in mice causes neutropenia and impair host defense against microbial infection (JEM, 206, 1181-99, 2009; AJP, 18, 1504-1512, 2012). In this report, the authors show that castrated male mice develop more lung colonies in experimental melanoma 'metastasis' models (i.e., tail vein-injected melanoma cells), a phenomenon that can be completely suppressed by T (testosterone) replacement. They further link this increased 'metastasis burden' to impaired neutrophil maturation as a result of loss of androgens. They also demonstrate that abnormal neutrophil phenotype in prostate cancer patients receiving ADT (androgen deprivation therapy). The study is interesting and sheds novel light on how androgen signaling may modulate immune control of cancers via regulating an innate immune component, i.e., neutrophil development and functions. Overall, studies were well designed, and data were solid supporting the conclusions.

Major points:

1. Do the authors have any direct evidence (e.g., EM or other functional assays) that mature neutrophils (or NK cells) in the male mouse lung actually engulf and kill injected melanoma cells and that this defense mechanism is impaired in the lungs of castrated male mice?
2. Along the same line, if male and female mice are treated with antiandrogens such as enzalutamide, do they show increased colonies in the lung upon inoculation of melanoma cells.
3. Images in Figure 1h should be replaced with better ones.
4. Qualification and quantification of (mature vs. immature) neutrophils were largely based on the nuclear pattern, i.e., segmented vs. banded, under the light microscope. However, from several representative images shown (e.g., Suppl. Fig. 3e; Suppl. Fig. 4h; Suppl. Fig. 5a), the "S" and "B" banding patterns frequently look indistinguishable. This casts doubts on the validity of this body of data.
5. Data on Profiler qPCR array of AR signaling target in neutrophils isolated from sham vs. castrated male mice (Suppl. Fig. 3h-i) is intriguing, as nearly 50% of 'AR-target' genes become upregulated in response to castration. This is a bit surprising as most AR targets are expected to go down in response to castration. Did the authors check on the AR signaling target in the mouse prostates?

Minor points:

1. What's point of showing reduced IL-1beta and increased IL-6 in female mice in (Supplementary Fig. 1c-d)?
2. A Western blot should be shown to illustrate the lack of AR in B16 melanoma cells (Supplementary Fig. 1j).
3. The reference "Kissick, H.T. et al. Androgens alter T-cell immunity by inhibiting T-helper 1 differentiation. Proc Natl Acad Sci U S A 111, 9887-9892 (2014)" should be cited.

Reviewer #2 (Remarks to the Author):

The manuscript by Markman and colleagues deals with the role played testosterone in regulating the function of neutrophils in the context of cancer. The basic observation that metastasis is more prevalent in older males has lead the authors to study how castration affects metastatic seeding of melanoma cells. The authors nicely show how castrated mice have increased metastatic load compared with control male which translates to reduced survival. The authors then use various

methods to implicate innate immune cells, and specifically neutrophils, in this change in the extent of metastatic colonization. Importantly, the authors show that it is testosterone which is critical for the activation of an anti-tumor phenotype in neutrophils and suggest that the reduced levels of testosterone may be the cause of increased metastatic spread in aging male melanoma patients. The manuscript is interesting, timely and well written. From a clinical perspective, the take home message from this study is that testosterone levels should be closely monitored and controlled in aging men to limit metastatic spread. However, several issues need to be resolved before this manuscript can be accepted for publication.

Major:

This is a well executed study, however, there is a significant caveat due to the experimental methodology used. The authors nicely show that the effect of testosterone on metastatic seeding using the experimental metastasis platform. They also show that primary tumor growth is unaffected by castration. Accordingly, the authors propose that neutrophil function at the future site of metastasis, but not at the primary site, is modulated by testosterone. However, this was never tested directly. In real life, metastatic seeding occurs when a primary tumor is present and has systemic effects including the generation of a premetastatic niche and modulation of the immune response. This is not modeled by the experimental metastasis platform which tests the seeding capacity of tail vein injected cells in an otherwise healthy mouse. The authors should show the consequences of castration (with and without testosterone replacement) affects spontaneous metastatic progression.

Minor:

In figure 3 the authors test the consequences of neutrophil depletion on metastatic seeding. The authors should show the extent of neutrophil depletion during the experiment.

Reviewer #3 (Remarks to the Author):

REVIEW: Loss of testosterone impairs anti-tumor neutrophil function

Major questions:

1. The fundamental question is how these data translate to humans. Melanoma primarily affects and kills older men, but lung metastases in humans, in contrast to the data presented here, are not known to be significantly more prevalent in females. To support the rationale of the paper, these human details should be audited and presented.
2. In principle, this paper aims to unravel the sex bias in outcome for mm patients, however it shows female mice have more metastases (although this is not a metastasis model but a tail vein model), and that in male animals, you can "restore" the metastatic capacity by castration to the level of females. These data do not explain why men have in fact more metastases, a shorter PFS and lower OS, but actually show the opposite trend. There is no stat sig difference in the number of male castrated to female mice (1f). thus, they cannot claim their data explains why men have a higher "incidence", as claimed in the discussion, or why they have a worse outcome.
3. These studies use a tail vein model with the monosomic b16 melanoma cell line. This is an extremely limited model to use and base such studies, as both the cell line and the model are not accepted to truly represent melanoma or a true metastasis model.
4. A monosomic cell line and a male cell line are injected into male/castrated/female animals. How do they control for immune rejection of and xy cell line in females, immune effects of a monosomic x cell line, and how would that not affect ntf function? The principle experiments should be repeated with multiple xx and xy cell lines to discern autoimmune effects, as the BMT experiment addresses the sex of the immune cell, but perhaps does not address the possibility of an observed

difference driven by an autoimmune response.

5. Neutrophils have been shown to actively contribute to melanoma and cancer metastasis, and to limit the response to novel immunotherapies (Karen de Visser, Thomas Tuting...). How do the seminal ntf studies from previous ntf-metastasis experts fit in with the findings in this paper, where loss in the percentage of neutrophils leads to metastasis?

6. This study assumes equal melanin production from the same cell line across different sites. B16 can dramatically vary its pigment production across animals, within the same anatomic tumour site, within the same experiment; so melanin content is not an accurate measure of mets, but histology is. And cell type is not shown in the supplementary figure 1. Moreover, these need to show ntf stains. Flow cytometry is a supportive experiment; the critical experiment is cell type in situ, and homogenization of the lung to investigate cell type a very limited approach.

Point-by-point responses and changes made to the manuscript.

Reviewers' comments:

Reviewer #1 (General Remarks to the Author):

“Androgens/androgen receptor signaling have a great influence on the development and functions of immune system. Previous studies in genetic mouse models have demonstrated that androgen overall induce immune-suppression and that Ar deficiency in mice causes neutropenia and impair host defense against microbial infection (JEM, 206, 1181-99, 2009; AJP, 18, 1504-1512, 2012). In this report, the authors show that castrated male mice develop more lung colonies in experimental melanoma ‘metastasis’ models (i.e., tail vein-injected melanoma cells), a phenomenon that can be completely suppressed by T (testosterone) replacement. They further link this increased ‘metastasis burden’ to impaired neutrophil maturation as a result of loss of androgens. They also demonstrate that abnormal neutrophil phenotype in prostate cancer patients receiving ADT (androgen deprivation therapy). The study is interesting and sheds novel light on how androgen signaling may modulate immune control of cancers via regulating an innate immune component, i.e., neutrophil development and functions. Overall, studies were well designed, and data were solid supporting the conclusions”.

We thank Reviewer #1 for their thorough and constructive comments. As described below, we believe that we have now addressed all of the critiques and suggestions from this reviewer, greatly strengthening our manuscript.

Major points:

Question 1. *“Do the authors have any direct evidence (e.g., EM or other functional assays) that mature neutrophils (or NK cells) in the male mouse lung actually engulf and kill injected melanoma cells and that this defense mechanism is impaired in the lungs of castrated male mice?”*

Response 1. While we do not have any live *in vivo* imaging in the lungs of mice, we have added additional experiments to attempt to address this question. Neutrophils and NK cells were isolated directly from the lungs of B16 tumor-bearing mice and were co-cultured with B16 cells to assess cytotoxicity. These experiments revealed a non-significant trend of decreased cytotoxicity by cells isolated from castrated male mice (**Fig. 2i-j**). Additionally, we now show that neutrophils from the lungs of B16 tumor-bearing mice are more phagocytic than neutrophils from castrated mice or female mice (**Fig. 3g**).

Question 2. *“Along the same line, if male and female mice are treated with antiandrogens such as enzalutamide, do they show increased colonies in the lung upon inoculation of melanoma cells”.*

Response 2. Thank you for this suggestion. We have added additional experiments treating mice with a flutamide pellet. Indeed, treatment with flutamide did significantly increase tumor burden in both male (**Fig. 4k-l, Supp. 4i**) and female mice (**Supp. 4j-k**).

Question 3. *“Images in Figure 1h should be replaced with better ones”.*

Response 3. Thank you for pointing this out to us. We have redone the figure using higher resolution images.

Question 4. *“Qualification and quantification of (mature vs. immature) neutrophils were largely based on the nuclear pattern, i.e., segmented vs. banded, under the light microscope. However, from several representative images shown (e.g., Suppl. Fig. 3e; Suppl. Fig. 4h; Suppl. Fig. 5a), the “S” and “B” banding patterns frequently look indistinguishable. This casts doubts on the validity of this body of data”.*

Response 4. Thank you for your comment. While at first glance distinguishing segmented from banded neutrophils may seem difficult, when viewed as a larger image, the neutrophil types are fairly easy to distinguish. Because some cells may be scored in between categories, we ensured that all samples were scored blindly by two investigators, and only scores that were in agreement between the two scientists were included. At least 150 neutrophils per sample were scored by each investigator. We used the below figure from Pillay, et al in “Immune suppression by neutrophils and granulocytic myeloid-derived suppressor cells: similarities and differences” (*Cell Mol Life Sci.* 2013 70(20)3813-27) as a guideline for scoring.

Question 5. “Data on Profiler qPCR array of AR signaling target in neutrophils isolated from sham vs. castrated male mice (Suppl. Fig. 3h-i) is intriguing, as nearly 50% of ‘AR-target’ genes become upregulated in response to castration. This is a bit surprising as most AR targets are expected to go down in response to castration. Did the authors check on the AR signaling target in the mouse prostates?”

Response 5. Thank you for bringing this to our attention. We have now gone back and extracted RNA from prostate lobes of both sham operated and castrated mice and performed the same AR target qPCR array. The results were nearly opposite of the neutrophil data, with 56 targets downregulated, and only 5 upregulated (>2-fold difference). We now mention these findings in the manuscript.

Minor points:

Minor Point 1. “What’s point of showing reduced IL-1beta and increased IL-6 in female mice in (Supplementary Fig. 1c-d)?”

Response 1. Thank you for this point. We agree that this does not add any mechanistic insight, and have removed the data.

Minor Point 2. “A Western blot should be shown to illustrate the lack of AR in B16 melanoma cells (Supplementary Fig. 1j).”

Thank you for this suggestion. While we did confirm this previously by western blot using the sc-815 antibody, the original image was lost due to a computer mishap. Furthermore, the sc-815 ab was discontinued by Santa Cruz. For the purpose of this response, we include below our lab notebook entry for the AR western blot, confirming B16 does not express AR. While this is obviously not optimal, in combination with our AR transcript data (**Supp. 1h**) we hope this is sufficient evidence that AR is not expressed in B16 cells.

Minor Point 3. “The reference “Kissick, H.T. et al. Androgens alter T-cell immunity by inhibiting T-helper 1 differentiation. *Proc Natl Acad Sci U S A* 111, 9887-9892 (2014)” should be cited”.

Response 3. Thank you for this reference, we have added it to our discussion. It certainly warrants further investigating Th1/2/17 differentiation in future experiments.

Reviewer #2 (General Remarks to the Author):

“The manuscript by Markman and colleagues deals with the role played testosterone in regulating the function of neutrophils in the context of cancer. The basic observation that metastasis is more prevalent in older males has led the authors to study how castration affects metastatic seeding of melanoma cells. The authors nicely show how castrated mice have increased metastatic load compared with control male which translates to reduced survival. The authors then use various methods to implicate innate immune cells, and specifically neutrophils, in this change in the extent of metastatic colonization. Importantly, the authors show that it is testosterone which is critical for the activation of an anti-tumor phenotype in neutrophils and suggest that the reduced levels of testosterone may be the cause of increase metastatic spread in aging male melanoma patients. The manuscript is interesting, timely and well written. From a clinical perspective, the take home message from this study is that testosterone levels should be closely monitored and controlled in aging men to limit metastatic spread. However, several issues need to be resolved before this manuscript can be accepted for publication”.

We thank reviewer 2 for their comments, which we fully agree with. We have addressed these concerns with new experiments, which have increased the impact of our study.

Major points:

Question1. “This is a well-executed study, however, there is a significant caveat due to the experimental methodology used. The authors nicely show that the effect of testosterone on metastatic seeding using the experimental metastasis platform. They also show that primary tumor growth is site of metastasis, but not at the primary site, is modulated by testosterone. However, this was never tested directly. In real life, metastatic seeding occurs when a primary tumor is present and has systemic effects including the generation of a premetastatic niche and modulation of the immune response. This is not modeled by the experimental metastasis platform which tests the seeding capacity of tail vein injected cells in an otherwise healthy mouse. The authors should show the consequences of castration (with and without testosterone replacement) affects spontaneous metastatic progression”.

Response 1. Thank you for these comments. While the B16 model was not suitable for this experiment due to low metastasis rate and fast primary tumor growth, the YUMM1.7 melanoma cells proved to be amenable for these studies. Thus, we have now assessed spontaneous metastasis with the YUMM1.7 melanoma cells in sham, castrated, castrated+T, and female mice. Consistent with the B16 cells, no difference was found between male and female mice in the primary tumors. However, we did find an increase in primary tumor size in castrated mice compared with sham and castrated plus testosterone mice (**Fig. 4d**). We also found that a significant increase in YUMM1.7 metastases in the lungs of castrated mice compared with sham and castrated plus testosterone mice (**Fig. 4e**). While the number of metastases was low, we feel that these new data strongly add support to our paper, and we thank the reviewer for this suggestion.

Minor point:

Minor point 1. *“In figure 3 the authors test the consequences of neutrophil depletion on metastatic seeding. The authors should show the extent of neutrophil depletion during the experiment”.*

Response 1. Thank you for this suggestion. We only checked the depletion efficacy at the end of the study (day 14-Supp Figure 3a), when we found the neutrophils were still depleted. Additionally, we felt that numerous blood draws would have introduced an additional experimental variable. However, prior to these studies while we were testing the efficacy of the Ly6G antibody, we did check blood, spleen, and lungs of mice at days 1, 3, and 5 and confirmed depletion (not shown). Finally, the 1A8 clone that we used for neutrophil depletion has been commonly used for neutrophil studies- such that they are too numerous to list.

Reviewer #3 (General Remarks to the Author):

We thank reviewer 3 for their comments. We have addressed these concerns with new experiments and more thorough explanations, which have added supportive data to our paper.

Major points:

Question 1. *“The fundamental question is how these data translate to humans. Melanoma primarily affects and kills older men, but lung metastases in humans, in contrast to the data presented here, are not known to be significantly more prevalent in females. To support the rationale of the paper, these human details should be audited and presented”.*

Response 1. Thank you. We have done a literature search to see if melanoma metastases occur in different sites/more frequently in certain sites like the lung in males versus females, but we were unable to find any publications that report on this (which we have now mentioned in the introduction). This would be very interesting to look at in the future.

Question 2. *“In principle, this paper aims to unravel the sex bias in outcome for mm patients, however it shows female mice have more metastases (although this is not a metastasis model but a tail vein model), and that in male animals, you can “restore“ the metastatic capacity by castration to the level of females. These data do not explain why men have in fact more metastases, a shorter PFS and lower OS, but actually show the opposite trend. **There is no stat sig difference in the number of male castrated to female mice (1f).** thus, they cannot claim their data explains why men have a higher “incidence “, as claimed in the discussion, or why they have a worse outcome”.*

Response 2. While we used melanoma as our model and thus based our discussion on this cancer, we mainly focused on the phenomenon of testosterone promoting the anti-tumor potential of neutrophils. We agree that we did not elucidate why males with melanoma have worse PFS and OS, and do not claim to. However, our data

does indicate that it may be interesting to look further at aging males, who have also had lower levels of testosterone, to see if hormones are exerting an effect on immune defense against cancer. Thus, our conclusion, as written in our discussion is: "In addition, while men and women initially have a similar incidence of melanoma, the risk rises in males dramatically after age 50⁴, correlating with an age-dependent drop in testosterone level⁵³ and in line with our observations in castrated mice. Thus, it may be beneficial to assess hormone levels in this subset of melanoma patients and elucidate the impact that neutrophils may have on the progression of cancer. Further studies to assess function of neutrophils in patients suffering from low T are also warranted."

Further, we have also added additional YUMM1.7 experiments in which spontaneous lung metastases developed from subcutaneous tumors (see response to reviewer 2 and **Fig. 4d-e**) at a higher rate in castrated male mice compared with sham male and castrated+T male mice.

Question 3. *"These studies use a tail vein model with the monosomic b16 melanoma cell line. This is an extremely limited model to use and base such studies, as both the cell line and the model are not accepted to truly represent melanoma or a true metastasis model"*

Response 3. We agree that tail vein injections are an experimental metastatic model, and while they are frequently used, there are limitations. We also understand that the B16 model has limitations in the reflection of human melanoma, which is why we also performed studies using the YUMM cells, which were derived from the GEMM mice containing common mutations found in human melanoma. As these data were consistent with the B16 data, we do not think our findings are limited only to B16 cells. Unfortunately, we were unable to show true metastasis from a primary B16 tumor to the lung as we experienced technical difficulties (see response to reviewer 2), but we have added data from a spontaneous model using the YUMM1.7 cells (**Fig. 4d-e**) to better represent a metastatic model.

Question 4. *"A monosomic cell line and a male cell line are injected into male/castrated/female animals. How do they control for immune rejection of and xy cell line in females, immune effects of a monosomic x cell line, and how would that not affect ntf function? The principle experiments should be repeated with multiple xx and xy cell lines to discern autoimmune effects, as the BMT experiment addresses the sex of the immune cell, but perhaps does not address the possibility of an observed difference driven by an autoimmune response"*

Response 4. Thank you for this suggestion. However, if immune rejection were occurring, we would expect female mice to exhibit a lower tumor burden, especially when XY cells were injected. As the female mice actually exhibited higher tumor burden when both an X and an XY cell line were injected, we do not believe immune rejection is an issue in our system. In addition, immune rejection is a function of the adaptive immune system, but the sex difference is still present in mice without adaptive immune cells (Rag1^{-/-} mice, Fig. 2D). Nonetheless, we repeated the study in male and female mice using an additional cell line derived from GEMM mice with human melanoma mutations (BRA^{V600E}wt/Cdkn2^{-/-}). This YUMM3.1 cell line was derived from a female mouse (XX) and exhibited the same increased tumor burden in female mice compared with male mice as the YUMM1.7 (XY) and B16 (X) melanoma lines (**Supp. 1i-j**).

Question 5. *"Neutrophils have been shown to actively contribute to melanoma and cancer metastasis, and to limit the response to novel immunotherapies (Karen de visser, Thomas tuting...). How do the seminal ntf studies from previous ntf-metastasis experts fit in with the findings in this paper, where loss in the percentage of neutrophils leads to metastasis?"*

Response 5. We are sorry for this confusion. We have expanded our discussion of the anti-tumor and pro-tumor roles that neutrophils have in various types of cancer (please see lines 320-348 in the manuscript) and have included reference to Karin de Visser's findings. While the potential impact of neutrophils on novel immunotherapies is very interesting and would likely require a combination of eliminating pro-tumor neutrophils while enhancing anti-tumor neutrophils, we find that a thorough discussion of this is beyond the scope of this paper and would be more suitable for a follow-up study.

Question 6. *“This study assumes equal melanin production from the same cell line across different sites. B16 can dramatically vary its pigment production across animals, within the same anatomic tumour site, within the same experiment; so melanin content is not an accurate measure of mets, but histology is. And cell type is not shown in the supplementary figure 1. Moreover, these need to show ntf stains. Flow cytometry is a supportive experiment; the critical experiment is cell type in situ, and homogenization of the lung to investigate cell type a very limited approach”.*

Response 6. We agree that melanin production from the B16 cells can vary, even within an anatomic site. Indeed, while counting the number of tumor colonies under magnification, we often see tumors that range from dark black to gray to white, which is why we report our findings as a tumor count, rather than by melanin content, throughout the study. However, we did extract melanin from many of our studies and did see that it always reflected tumor counts, which is shown in Supp 1A. Our main purpose for showing this was because we needed to extract melanin from the NK1.1 depleted mice (Supp 3c-d) as the tumor burden was too high to be accurately counted. We also show representative histology staining in Supp 1b, which we confirmed reflects tumor count in multiple studies.

Thank you for bringing to our attention that we did not denote B16 in Supp 1a, we have revised the figure legend to correct this.

We now also performed IHC for neutrophils in lungs of B16-tumor bearing mice. We did find more neutrophils in contact with or closely located to the tumor in sham operated male mice compared with castrated male or female mice (**Fig. 3h**). As there were not a large number of neutrophils surrounding the tumor area, we quantified the distance of neutrophils within 100 μm from the tumor surface.

Reviewers' comments:

Reviewer #1 (Remarks to the Author):

In this paper, Markman et al demonstrate that "loss of testosterone impairs anti-tumor neutrophil function". Important messages are embedded in this study that has relevance in understanding the cancer biology behind increased incidence of certain cancers in aging male population (presumably related to decreasing levels of androgens) and how clinically used castration regimens and anti-androgens might actually promote tumor progression by negatively impacting innate immune cell functions. In this revised, improved manuscript, the authors made efforts to address some of my questions but there still exist some technical issues that preclude firm conclusions.

1. Some data related to the YUMM1.7 model are problematic. Images shown in Fig. 1h and Supplementary Fig. 1i are simply too dark to be informative, making one wonder how the authors derived the colony numbers shown in Fig. 1i and Supplementary Fig. 1j. Cross-section HE images similar to Supplementary Fig. 1b should be presented. Also, as I pointed out earlier, differential AR expression in B16 vs. YUMM1.7 cells needs to be better characterized and western blotting data should be presented.

2. Related to the point of "lung colony quantification": in the B16 system, there often exist significant discrepancies in the numbers presented, e.g., ~100-200 lung colonies in males in Fig. 1b vs. ~10 lung colonies in 'male+sham' group in Fig. 4b. Similar issues seem to exist for YUMM1.7. For example, Fig. 1i shows experimental lung colonies to be ~25 and 60 in 'Male+sham' and 'Male+castration' groups, respectively. In the 'spontaneous' metastasis model, these two groups of mice showed only ~1 vs. 3 lung colonies (Fig. 4e). Overall, these discrepancies again cast doubt in the reproducibility and validity of various experiments and robustness of data.

3. To help readers more easily navigate dense data and busy figure panels, the individual immune cell types analyzed in Supplementary Fig. 2a should be clearly labeled.

4. There are many typographical and grammatical errors.

5. I previously commented "Data on Profiler qPCR array of AR signaling target in neutrophils isolated from sham vs. castrated male mice (Suppl. Fig. 3h-i) is intriguing, as nearly 50% of 'AR-target' genes become upregulated in response to castration. This is a bit surprising as most AR targets are expected to go down in response to castration. Did the authors check on the AR signaling target in the mouse prostates?" Now the authors did the suggested experiments and indeed, obtained contrasting expression patterns in AR target genes in castrated prostates but, puzzlingly, did not present the data (page 8-9). These data are critical and should be presented (with replicate biological samples and statistical analysis).

6. Authors claim that ".....neutrophils isolated from the bone marrow of tumor bearing castrated male mice expressed lower CXCR2 mRNA levels and higher CXCR4 and VLA-4 mRNA levels compared with those from sham male mice (Supplementary Fig. 3g)" but there were no statistics to support these statements.

7. Authors made the comment that "..... T may be beneficial to the immune response at physiologic levels, but detrimental at super-physiologic doses" (P10, bottom). This has important implications for BAT (Bipolar Androgen Therapy) currently in trials in prostate cancer patients. Authors might want to comment on this point in Discussion.

Reviewer #2 (Remarks to the Author):

The authors adequately addressed my concerns.

Reviewer #3 (Remarks to the Author):

Overall, the interaction between neutrophils, NK cells, the AR pathways is intriguing and unexplored in melanoma and other cancers. The work depicted here convincingly shows that in animals, the hormonal androgen status influences NTF function, which will affect tumour progression. The experiments are well conducted. The addition of the YUMM cell line work is valuable. It is a valuable contribution but I question the wider scientific and clinical context.

Three fundamental questions remain.

1. Women present a delayed course of melanoma metastasis and less frequent mets in the lung, in favour of locoregional disease. The fact their data and study hinges on a mouse model that exactly shows the opposite is problematic and questions the ultimate value of the work. It does not model a real clinical scenario. More importantly, the question of anatomic site metastasis by sex could be easily audited, and some of the data are published, despite the authors declaring these data are unpublished. Evidence of female lung disease vs male lung disease are published observing the contrary of their findings in their mouse model of tail vein injection (response to reviewer 3, point 1): PLoS One. 2012; 7(3): e32955. Time Course and Pattern of Metastasis of Cutaneous Melanoma Differ between Men and Women, and refs therein.

2. The second point is that regardless of NTF N1 vs N2, pro tumour vs anti tumour effects, the current human data shows that mature, segmented NTF infiltrating primary melanoma human samples are associated with a worse outcome (Suppl material in Nature, 2014, Thomas Tuting). The authors have now added to the discussion the data on pro-met vs anti-met of NTF across human cancers but fail to discuss the critical point: in human melanoma, NTF infiltrates in primary disease are a negative predictor of outcome. I appreciate these human data do not cancel the mouse and in vivo work in this study, but surely limit the significance/ interpretation of the study and are not appropriately discussed.

3. Do the findings in NTF of peripheral blood of patients with/without ADT prostate treatment arise from multivariate analysis accounting for total WBC count, and more importantly age?

Point-by-point responses and changes made to the manuscript.

Reviewer #1 (Remarks to the Author):

General Comment: *“In this paper, Markman et al demonstrate that ‘loss of testosterone impairs anti-tumor neutrophil function’”. Important messages are embedded in this study that has relevance in understanding the cancer biology behind increased incidence of certain cancers in aging male population (presumably related to decreasing levels of androgens) and how clinically used castration regimens and anti-androgens might actually promote tumor progression by negatively impacting innate immune cell functions. In this revised, improved manuscript, the authors made efforts to address some of my questions but there still exist some technical issues that preclude firm conclusions”.*

Response: Thank you for your careful reading and helpful suggestions and the enthusiastic comments about the important messages in our improved revised manuscript.

Remaining few Comments and our responses and changes we made are listed below:

Comment 1. “Some data related to the YUMM1.7 model are problematic. Images shown in Fig. 1h and Supplementary Fig. 1i are simply too dark to be informative, making one wonder how the authors derived the colony numbers shown in Fig. 1i and Supplementary Fig. 1j. Cross-section HE images similar to Supplementary Fig. 1b should be presented. Also, as I pointed out earlier, differential AR expression in B16 vs. YUMM1.7 cells needs to be better characterized and western blotting data should be presented”.

Response 1: It is indeed difficult to photograph the lungs after they are dyed and fixed with India Ink. However, we counted white colonies using a microscope and a light source that made them easily visible. In addition, a second blinded researcher confirmed the counts with an accuracy of +/- 0-2 colonies in the YUMM1.7 model (and +/- 0-5 for B16 cells), ensuring that our method and results were robust. This technique is an accepted method of analysis used in multiple published studies (Miretti, et al., 2008 *PLoS One*, 3(3):e1928; Zimmerman, et al., 2010, *J. Vis Exp*, 45:2077; Dong, et al. 2015. *J Vis Exp* 99:52609; etc.). As the lungs are fixed and permanently stained during this process, we are unable to re-analyze the tissue with H&E stain. We feel that the images presented clearly show the differences in tumor burden among these groups (as evidenced by increased white areas indicating increased tumor burden), and thus we do not think it is justifiable to repeat this 7 week experiment (4 weeks post castration, plus 3 weeks for tumor growth) and sacrifice more mice for this purpose, as the first round of revisions already required over 100 additional animals.

As requested, we have now added western blot data for both the B16 and YUMM1.7 cells (Supplementary Fig. 1i), and added comments in the main text (lines 98-100) as well at the methods. As both cell lines appear negative for the androgen receptor protein, the addition of these experiments did not change any of our conclusions.

Comment 2. *“Related to the point of ‘lung colony quantification’: in the B16 system, there often exist significant discrepancies in the numbers presented, e.g., ~100-200 lung colonies in males in Fig. 1b vs. ~10 lung colonies in ‘male+sham’ group in Fig. 4b. Similar issues seem to exist for YUMM1.7. For example, Fig. 1i shows experimental lung colonies to be ~25 and 60 in ‘Male+sham’ and ‘Male+castration’ groups, respectively. In the ‘spontaneous’ metastasis model, these two groups of mice showed only ~1 vs. 3 lung colonies (Fig. 4e). Overall, these discrepancies again cast doubt in the reproducibility and validity of various experiments and robustness of data”.*

Response 2: In these experiments as in all biological research, we observed some variability between replicates. However, we believe the reviewer may have misread the graph axes and misinterpreted

some of the results. For example, in Figure 1b, the average number of colonies in the male+sham group was 126, and in Figure 4b, the average was 92. The trends, e.g. increased tumor burden in castrated male and female mice compared with sham-operated male mice, remain very consistent throughout all the experiments. We did not transform or normalize any of our data, and present the individual data points rather than bar graphs, to enhance transparency. Throughout these experiments, we were very careful to inject the cells at the same passage number, the same number of cells, and during the same time of day.

Regarding the “spontaneous” model having fewer metastases than the tail vein injection model, this is not surprising, given that they are independent models. We do believe that the number of spontaneous tumors would increase with time, however, the size of the primary tumors caused the mice to reach euthanasia criteria quickly, terminating the experiment.

Comment 3. *“To help readers more easily navigate dense data and busy figure panels, the individual immune cell types analyzed in Supplementary Fig. 2a should be clearly labeled”.*

Response 3: Thank you for this suggestion. We agree, and have labeled the figure more clearly in Supp. Fig. 2a.

Comment 4. *“There are many typographical and grammatical errors”.*

Response 4: Thank you. We have attempted to identify and correct any errors.

Comment 5. *“I previously commented “Data on Profiler qPCR array of AR signaling target in neutrophils isolated from sham vs. castrated male mice (Suppl. Fig. 3h-i) is intriguing, as nearly 50% of ‘AR-target’ genes become upregulated in response to castration. This is a bit surprising as most AR targets are expected to go down in response to castration. Did the authors check on the AR signaling target in the mouse prostates?” Now the authors did the suggested experiments and indeed, obtained contrasting expression patterns in AR target genes in castrated prostates but, puzzlingly, did not present the data (page 8-9). These data are critical and should be presented (with replicate biological samples and statistical analysis)”.*

Response 5: We have now added in the data from the prostate samples in **Supplementary 3i**. Castration results in a great reduction in size of the seminal vesicles and prostate lobes, making them extremely difficult to isolate, and severely limits the amount of tissue that can be obtained from an individual mouse. Therefore, we pooled the samples to obtain enough RNA to run the array, and do not have a sufficient number of samples for statistical analysis. As the purpose of this experiment was to show that the androgen receptor pathway was perturbed in neutrophils (as shown and validated in **Supplementary 3h and 3j**), we do not think it is necessary to repeat this experiment for replicate biological samples and statistical analysis of the prostate samples. Instead, we erred on the side of caution and only reported genes with greater than a 2-fold change.

Comment 6. *“Authors claim that “.....neutrophils isolated from the bone marrow of tumor bearing castrated male mice expressed lower CXCR2 mRNA levels and higher CXCR4 and VLA-4 mRNA levels compared with those from sham male mice (Supplementary Fig. 3g)” but there were no statistics to support these statements”.*

Response 6: The data in **Supplementary 3g** are calculated fold changes, which are not statistically analyzable (as far as we know). Thus, we do not claim any “statistical significance” in the text of the manuscript, and only report on increases and decreases in expression in castrated male or female mice compared with sham male mice.

Comment 7. *“Authors made the comment that “..... T may be beneficial to the immune response at physiologic levels, but detrimental at super-physiologic doses” (P10, bottom). This has important implications for BAT (Bipolar Androgen Therapy) currently in trials in prostate cancer patients. Authors might want to comment on this point in Discussion”.*

Response 7: Thank you for this suggestion. In this mouse model, the super-physiological T dose resulted in an average 8-fold higher level of circulating testosterone. In contrast, Tepley et al. (Lancet Oncol. 2018 Jan; 19(1): 76–86) reported that after 3 cycles of BAT, nadir median total testosterone concentrations increased from 20 ng/dL to 207.5 ng/dL, which would NOT constitute a supraphysiologic concentration. This is an extremely controversial form of treatment causing great excitement in the field, but likely is not directly related to this work. While we appreciate the suggestion, we would prefer to avoid the controversies of this therapy in this discussion.

Reviewer #2 (Remarks to the Author):

The authors adequately addressed my concerns.

Thank you for your thoughtful review of our revised manuscript.

Reviewer #3 (Remarks to the Author):

General Comments: *“Overall, the interaction between neutrophils, NK cells, the AR pathways is intriguing and unexplored in melanoma and other cancers. The work depicted here convincingly shows that in animals, the hormonal androgen status influences NTF function, which will affect tumour progression. The experiments are well conducted. The addition of the YUMM cell line work is valuable. It is a valuable contribution but I question the wider scientific and clinical context”.*

Response: Thank you for these strongly positive comments. We hope we have now addressed your few remaining questions satisfactorily as shown below.

Three fundamental questions remain.

Comment 1. *“Women present a delayed course of melanoma metastasis and less frequent mets in the lung, in favour of locoregional disease. The fact their data and study hinges on a mouse model that exactly shows the opposite is problematic and questions the ultimate value of the work. It does not model a real clinical scenario. More importantly, the question of anatomic site metastasis by sex could be easily audited, and some of the data are published, despite the authors declaring these data are unpublished. Evidence of female lung disease vs male lung disease are published observing the contrary of their findings in their mouse model of tail vein injection (response to reviewer 3, point 1): PLoS One. 2012; 7(3): e32955. Time Course and Pattern of Metastasis of Cutaneous Melanoma Differ between Men and Women, and refs therein”.*

Response 1: Thank you for these comments and the paper reference, which we missed in our previous literature search. We have incorporated this reference into the manuscript (lines 40-43) in reference to locoregional versus distant metastasis in males and females.

The main focus of our manuscript is to understand the critical role of androgen signaling in neutrophil function, and the potential impact of this effect on cancer progression.” We agree that our data do not explain why males with melanoma have worse PFS and OS, and do not claim to have solved this puzzle. However, our data do indicate that it may be interesting to look further at **aging males**, who have lower levels of testosterone, to see if hormones exert an effect on immune defense against cancer. Indeed, in our discussion we pointed out that the risk of melanoma rises dramatically after age 50 in males, which correlates with an age dependent drop in testosterone levels.

Comment 2. *“The second point is that regardless of NTF N1 vs N2, pro tumour vs anti tumour effects, the current human data shows that mature, segmented NTF infiltrating primary melanoma human samples are associated with a worse outcome (Suppl material in Nature, 2014, Thomas Tuting). The authors have now added to the discussion the data on pro-met vs anti-met of NTF across human cancers but fail to discuss the critical point: in human melanoma, NTF infiltrates in primary disease are a negative predictor of outcome. I appreciate these human data do not cancel the mouse and in vivo work in this study, but surely limit the significance/ interpretation of the study and are not appropriately discussed”.*

Response 2 : Thank you for this comment. We have expanded our discussion on the pro-tumor role of neutrophils, and added more examples specifically for melanoma (lines 328-334). As the reviewer notes, the paper by Thomas Tuting, which we now cite, states, “In an unselected sentinel node-staged patient cohort treated at our institution we verified that superficial ulceration and prominent neutrophil infiltration of primary melanomas is associated with signs of angiotropism and lymph node metastasis (Extended Data Fig. 9).” However, the actual figure does not show any specific information on neutrophil infiltration, but rather focuses on presence of angiotropism and ulceration, and includes staining for CD31 only. In addition, the Tuting paper does not include evidence that the infiltrating neutrophils are mature segmented neutrophils. Finally, the neutrophils described are in ulcerated tumors, so the cause of their localization is unknown, and may be related to healing or possible infection. Therefore, we respectfully do not agree that the findings in the Tuting manuscript are in conflict with our data.

Comment 3. *“Do the findings in NTF of peripheral blood of patients with/without ADT prostate treatment arise from multivariate analysis accounting for total WBC count, and more importantly age?”*

Response 3: Thank you for this interesting question. Age was included in this multivariate analysis. As stated in the “Statistical Analysis” methods section: “For patient data, a multivariable linear regression analysis with adjustments for presence of prostatectomy, **age**, race and current status was performed by the Biostatistics Core at Cedars Sinai.” However, this IRB was part of a larger study in which we only received the neutrophils and red blood cell pellet, so we were unable to look at WBCs. Unfortunately, WBCs were not analyzed by our collaborators either, so we were not able to include this data in our analysis. We have now directly stated this in the methods section to make it clearer for the readers.

REVIEWERS' COMMENTS:

Reviewer #1 (Remarks to the Author):

The authors have made efforts to adequately address most of my questions. However, about my 'Comment 2', the authors responded, in the rebuttal letter, that 'For example, in Figure 1b, the average number of colonies in the male+sham group was 16, and in Figure 4b, the average was 92.' This seems incorrect: Figure 1b shows up to ~200 B16 lung colonies in the 'male' group whereas Figure 4b shows ~16 lung colonies in 'male+sham' group and a wide range of colonies in the 'Male+castration' group. Please clarify.

Reviewer #3 (Remarks to the Author):

1. The authors state: "The main focus of our manuscript is to understand the critical role of androgen signaling in neutrophil function, and the potential impact of this effect on cancer progression. We agree that our data do not explain why males with melanoma have worse PFS and OS, and do not claim to have solved this puzzle. However, our data do indicate that it may be interesting to look further at aging males, who have lower levels of testosterone, to see if hormones exert an effect on immune defense against cancer." However, the main focus of the paper does follow from an observation that is opposite to the human scenario. One would expect that lowering the levels of testosterone in aging males would lead to increased progression, as observed in the clinic. Therefore, they do not only "not solve the puzzle", but present a model and base all their work on the opposite premise.
2. The authors do not accept Tuting's data strongly suggest mature neutrophils (they are mature and segmented as can be seen across all pathological specimens, in this reviewer's opinion) are detrimental to melanoma outcome. There is however additional evidence to support mature neutrophil infiltration of melanoma will lead to a poor prognosis: For example: Intratumoral neutrophils and plasmacytoid dendritic cells indicate poor prognosis and are associated with pSTAT3 expression in AJCC stage I/II melanoma, Jensen et al.
3. NTF count in peripheral blood are also a predictor of poor outcome, and should be accounted for as absolute in proportion to total WBC to draw any conclusions.

Response to Reviewers

Reviewer #1

The authors have made efforts to adequately address most of my questions. However, about my 'Comment 2', the authors responded, in the rebuttal letter, that 'For example, in Figure 1b, the average number of colonies in the male+sham group was 16, and in Figure 4b, the average was 92.' This seems incorrect: Figure 1b shows up to ~200 B16 lung colonies in the 'male' group whereas Figure 4b shows ~16 lung colonies in 'male+sham' group and a wide range of colonies in the 'Male+castration' group. Please clarify.

Response:

We thank the reviewer for their question. We believe that there was a typo in the response as it should have read "the average number of colonies in the male+sham group was 126", not 16. Additionally, while there is variability between experiments, we were very careful to inject the cells at the same passage number, the same number of cells, and during the same time of day. However, despite these precautions, the results of each individual study did vary in the number of tumors. However, in each study we performed, all the trends and significant differences observed were uniform. Thus, while there was biological variability between individual experiments, the conclusions of each experiment were the same.

Reviewer #3

1. The authors state: "The main focus of our manuscript is to understand the critical role of androgen signaling in neutrophil function, and the potential impact of this effect on cancer progression. We agree that our data do not explain why males with melanoma have worse PFS and OS, and do not claim to have solved this puzzle. However, our data do indicate that it may be interesting to look further at aging males, who have lower levels of testosterone, to see if hormones exert an effect on immune defense against cancer." However, the main focus of the paper does follow from an observation that is opposite to the human scenario. One would expect that lowering the levels of testosterone in aging males would lead to increased progression, as observed in the clinic. Therefore, they do not only "not solve the puzzle", but present a model and base all their work on the opposite premise.

Response:

We have now clarified this in our study in lines 430-433 of our revised manuscript.

2. The authors do not accept Tuting's data strongly suggest mature neutrophils (they are mature and segmented as can be seen across all pathological specimens, in this reviewer's opinion) are detrimental to melanoma outcome. There is however additional evidence to

support mature neutrophil infiltration of melanoma will lead to a poor prognosis: For example: Intratumoral neutrophils and plasmacytoid dendritic cells indicate poor prognosis and are associated with pSTAT3 expression in AJCC stage I/II melanoma, Jensen et al.

Response:

We have now added these two references to our discussion (lines 332-333).

3. NTF count in peripheral blood are also a predictor of poor outcome, and should be accounted for as absolute in proportion to total WBC to draw any conclusions.

Response:

Unfortunately, per our IRB, we did not have access to WBC for our human data.